# Solving Zero-Sum Markov Games with Continuous State via Spectral Dynamic Embedding

**Chenhao Zhou**[1]  **Zebang Shen**[2]  **Chao Zhang**[1]*  **Hanbin Zhao**[1]  **Hui Qian**[1,3]

[1]College of Computer Science and Technology, Zhejiang University
[2]Department of Computer Science, ETH Zurich
[3]State Key Lab of CAD&CG, Zhejiang University
`{zhouchenhao,zczju,zhaohanbin,qianhui}@zju.edu.cn`
`zebang.shen@inf.ethz.ch`

## Abstract

In this paper, we propose a provably efficient natural policy gradient algorithm called Spectral Dynamic Embedding Policy Optimization (`SDEPO`) for two-player zero-sum stochastic Markov games with continuous state space and finite action space. In the policy evaluation procedure of our algorithm, a novel kernel embedding method is employed to construct a finite-dimensional linear approximations to the state-action value function. We explicitly analyze the approximation error in policy evaluation, and show that `SDEPO` achieves an $\tilde{O}(\frac{1}{(1-\gamma)^3\epsilon})$ last-iterate convergence to the $\epsilon-$optimal Nash equilibrium, which is independent of the cardinality of the state space. The complexity result matches the best-known results for global convergence of policy gradient algorithms for single agent setting. Moreover, we also propose a practical variant of `SDEPO` to deal with continuous action space and empirical results demonstrate the practical superiority of the proposed method.

## 1   Introduction

Two-player zero-sum stochastic Markov games (`2p0s-MGs`) has been the focus of research across a range of research communities. In this problem, two players select their actions based on the current state simultaneously and independently. Player one aims to maximize the return based on the reward provided by the environment, while player two aims to minimize it. For `2p0s-MGs` with finite state space, tabular methods [Alacaoglu et al., 2022, Bai and Jin, 2020, Daskalakis et al., 2020, Wei et al., 2021, Zhao et al., 2022] represent the state-action value function with tables, which results in a sample complexity depending on the cardinality of the state spaces.

To deal with `2p0s-MGs` with complex state space, researchers recently employ function approximations of the state-action value function to deal with large-scale discrete/continuous state space, including linear function approximations [Xie et al., 2020, Chen et al., 2022], kernel function approximations [Junchi Li et al., 2022, Qiu et al., 2021] and general function classes such as neural networks [Jin et al., 2022, Huang et al., 2021]. Basically, these methods use samples to construct the function approximations and update the policies with value iteration. Theoretical analyses show that these methods possess sample complexities independent of the state space's cardinality. Note that these methods fail to explicitly utilize the dynamics information of the underlying environments and only achieve a sub-optimal sample complexity $\tilde{O}(\epsilon^{-2})$ to find an $\epsilon-$optimal Nash equilibrium for `2p0s-MGs` with known system dynamics.

In this paper, we introduce a spectral dynamic embedding method, which explicitly uses the dynamics information to approximate the state-action value functions, and propose an efficient natural-policy-gradient-type algorithm, called Spectral Dynamic Embedding Policy optimization (`SDEPO`),

---

*Corresponding author.

| Dynamic | State Space | Reference | Iteration Complexity | Last-iterate Convergence | Horizon Length |
|---|---|---|---|---|---|
| Unknown | Finite | **Alacaoglu et al. [2022]** | $\widetilde{O}\left(\frac{1}{(1-\gamma)^3\epsilon}\right)$ | Yes | Infinite |
| | | **Zhao et al. [2022]** | $\widetilde{O}\left(\frac{1}{(1-\gamma)^3\epsilon}\right)$ | Yes | Infinite |
| | Infinite | **Xie et al. [2020]** | $\widetilde{O}\left(\frac{d^{3/2}H^2}{\epsilon^2}\right)$ | No | Finite |
| | | **Chen et al. [2022]** | $\widetilde{O}\left(\frac{dH^{3/2}}{\epsilon^2}\right)$ | No | Finite |
| | | **Jin et al. [2022]** | $\widetilde{O}\left(\frac{dH^2}{\epsilon^2}\right)$ | No | Finite |
| | | **Huang et al. [2021]** | $\widetilde{O}\left(\frac{dH^2}{\epsilon^2}\right)$ | No | Finite |
| | | **Qiu et al. [2021]** | $\widetilde{O}\left(\frac{H^2}{\epsilon^2}\right)$ | No | Finite |
| | | **Junchi Li et al. [2022]** | $\widetilde{O}\left(\frac{H^{3/2}}{\epsilon^2}\right)$ | Yes | Finite |
| Known | Finite | **Cen et al. [2021]** | $\widetilde{O}\left(\frac{1}{(1-\gamma)^3\epsilon}\right)$ | Yes | Finite |
| | | **Wei et al. [2021]** | $\widetilde{O}\left(\frac{|S|^3}{(1-\gamma)^9\epsilon^2}\right)$ | Yes | Infinite |
| | | **Zhang et al. [2022]** | $\widetilde{O}\left(\frac{H^4}{\epsilon}\right)$ | Yes | Finite |
| | Infinite | **This Work** | $\widetilde{O}\left(\frac{1}{(1-\gamma)^3\epsilon}\right)$ | **Yes** | Infinite |

Table 1: Comparison of policy optimization methods for finding an $\epsilon$-optimal NE of two-player zero-sum episodic Markov games in terms of the duality gap. Here, $H$ refers to the horizon length and $d$ is the dimension of their features. For simplicity, we ignore the problem-dependent constant.

for 2p0s-MGs. In particular, the spectral dynamic embedding method directly constructs truncated linear approximations to the transition dynamics of a Markov game in a kernel space, and implements dynamic programming to calculate the state-action value function approximation. The superiority of spectral dynamic embedding has been justified in single agent setting [Ren et al., 2022, 2023]. We leverage two kernel feature generation methods for the truncated approximation, namely random feature generation and Nyström feature generation, and analyze the approximation error of these two methods during policy evaluation. Our contributions lie in the following folds.

1. We present a truncated kernel-based linearization method for the state-action value function approximation in two-player zero-sum Markov games with continuous state space. With the random/Nyström feature generation, this method automatically generates truncated kernel representation from system dynamics, bypassing the difficulty of kernel feature decision existed in other kernel approximation methods. By integrating the acquired kernel features into the temporal difference learning process, we estimate the state-action value functions through the least square policy evaluation. Leveraging the obtained value functions, policy improvement can be achieved through the natural policy gradient descent/ascent approach.

2. We establish rigorous analysis of the approximation error in the truncated value-function approximation and the statistical error induced in policy improvement procedure with finite samples. Our theoretical analysis demonstrates that SDEPO achieves a near-optimal $\tilde{O}\left(\frac{1}{(1-\gamma)^3\epsilon}\right)$ last-iterate convergence to the $\epsilon$-optimal Nash equilibrium for 2p0s-MGs with continuous state space and finite action space, where $\gamma$ represents the discounted factor. This complexity result matches the best-known results for the policy gradient algorithm to achieve global convergence in the single agent setting.

Moreover, we also propose a practical variant of SDEPO to deal with continuous action space and empirical results demonstrate the practical superior performance of the proposed method.

## 1.1 Related works

**RL methods for 2p0s-MGs.** There is a large body of literature on MARL for two-player MGs. Alacaoglu et al. [2022], Daskalakis et al. [2020], Zhao et al. [2022] focus on the tabular setting, i.e., the state-action space can be represented by a table with moderate size To address the challenge of continuous state spaces, many researchers have designed algorithms based on function approximation. Xie et al. [2020], Chen et al. [2022] investigated methods based on linear function approximation, using a set of linear features to represent the state transition function and reward function. Jin et al. [2022], Huang et al. [2021] employed general function approximation for MGs with low multi-agent Bellman eluder dimension and MGs with a finite minimax Eluder dimension. Although

these algorithms yield strong theoretical guarantees, they are computationally inefficient. Junchi Li et al. [2022], Qiu et al. [2021] studied learning MGs with kernel approximation. Their approachs assume that there are a set of (possibly infinite-dimensional) kernel features that span the transition function or value function space. However, finding a good set of kernel features is a very challenging task, making their assumption difficult to satisfy. Additionally, infinite-dimensional kernel features are infeasible to compute, so their method requires finite-dimensional approximation, and the errors caused by finite-dimensional approximation currently lack analysis.

**Tabular methods for `2p0s-MGs` where system dynamics are known.** As shown in Table 1, a parallel line of research aims to solve `2p0s-MGs` with system dynamics known. For the infinite-horizon discounted setting, Wei et al. [2021] proposed an optimistic gradient descent ascent (OGDA) method which achieves a last-iterate convergence at an $\widetilde{O}\big(\frac{|S|^3}{(1-\gamma)^9\epsilon^2}\big)$ iteration complexity. Cen et al. [2021] established linear last-iterate convergence of entropy-regularized OMWU. For the finite-horizon episodic setting, Zhang et al. [2022] showed that the modified optimistic Follow-The-Regularized-Leader method finds an $\epsilon$-optimal NE in $\widetilde{O}\big(\frac{H^4}{\epsilon}\big)$ iterations and Cen et al. [2023] proposed a single-loop policy optimization algorithm that implies the last-iterate convergence with an iteration complexity of $\widetilde{O}\big(\frac{H^3}{\epsilon}\big)$. However, their methods are all confined to tabular setting, whereas our method can handle problems in continuous state spaces.

**RL with Function Approximation.** Function approximation in single-agent RL has been extensively studied in recent years to achieve a better sample complexity that depends on the complexity of function approximators rather than the size of the state/action space. One line of work studies RL with linear function approximation [Yang and Wang, 2019, Jin et al., 2020]. Typically, these methods assume the optimal value function can be well approximated by linear functions, and achieve polynomial sample efficiency guarantees related to feature dimension under certain regularity conditions. Another line of works studied the MDPs with general nonlinear function approximations [Jiang et al., 2017, Jin et al., 2021]. Jiang et al. [2017], Jin et al. [2021] present algorithms with PAC guarantees for problems with low Bellman rank and low BE dimension, respectively. We note that MGs are inherently more complex than MDPs due to their min-max nature and it is generally difficult to directly extend these results to the dual-player dynamic setting of MGs.

## 2  Background and Preliminaries

In this section, we present the necessary definitions that will be adopted throughout the paper. In Section 2.1, we formally describe the setup for two-player zero-sum stochastic Markov games with simultaneous moves. In Section 2.2, We briefly introduce the background knowledge about positive definite kernels and their decompositions.

### 2.1  Simultaneous-Move Markov Games

A two-player zero-sum stochastic Markov games with simultaneous moves is defined by the tuple $(\mathcal{S}, \mathcal{A}_1, \mathcal{A}_2, r, \mathbb{P}, \gamma)$, where $\mathcal{S}$ is the state space, $\mathcal{A}_i$ is the set of actions that player $i \in \{1, 2\}$ can take, $r$ is the reward function, $\mathbb{P}$ is the transition function and $\gamma \in [0, 1)$ is the discounted factor. At each step $t$, given the state $s \in \mathcal{S}$, P1 and P2 take actions $a \in \mathcal{A}_1$ and $b \in \mathcal{A}_2$, respectively, and then both receive the reward $r(s, a, b)$. The system then shifts to a new state $s' \sim \mathbb{P}(\cdot|s, a, b)$ according to the transition kernel. Throughout this paper, we assume for simplicity that $\mathcal{S} = \mathbb{R}^d$, $\mathcal{A}_1 = \mathcal{A}_2 = \mathcal{A}$ and that the rewards $r(s, a, b)$ are deterministic functions of the tuple $(s, a, b)$ taking value in $[-1, 1]$; generalization to the setting with $\mathcal{A}_1 \neq \mathcal{A}_2$ and stochastic rewards is straightforward.

Denote by $\Delta \equiv \Delta(\mathcal{A})$ the probability simplex over the action space $\mathcal{A}$. A stochastic policy of P1 is a sequence of functions $\overline{\pi} := (\overline{\pi}_t : \mathcal{S} \to \Delta)_t$. At each step $t$ and state $s \in \mathcal{S}$, P1 takes an action sampled from the distribution $\overline{\pi}_t(s)$ over $\mathcal{A}$. Similarly, a stochastic policy of P2 is given by the sequence $\underline{\pi} := (\underline{\pi}_t : \mathcal{S} \to \Delta)_t$.

For a fixed pair of policies $(\overline{\pi}, \underline{\pi})$ for both players, the value and Q (a.k.a. action-value) functions for the above game can be defined as following:

$$V^{\overline{\pi}, \underline{\pi}}(s) := \mathbb{E}\left[\sum_{t=0}^{\infty} \gamma^t r(s_t, a_t, b_t)|s_0 = s\right],$$

$$Q^{\overline{\pi}, \underline{\pi}}(s, a, b) := \mathbb{E}\left[\sum_{t=0}^{\infty} \gamma^t r(s_t, a_t, b_t)|s_0 = s, a_0 = a, b_0 = b\right],$$

where the expectation is over $a_t \sim \overline{\pi}_t(\cdot|s_t)$, $b_t \sim \underline{\pi}_t(\cdot|s_t)$ and $s_{t+1} \sim \mathbb{P}(\cdot|s_t, a_t, b_t)$.

In the zero-sum setting, for a given initial state $s_0$, P1 seeks to maximize $V^{\overline{\pi},\underline{\pi}}(s_0)$ whereas P2 aims to minimize it. Accordingly, we introduce the value and Q functions when P1 plays the best response to a fixed policy $\underline{\pi}$ of P2: $V^{*,\underline{\pi}}(s) = \max_{\overline{\pi}} V^{\overline{\pi},\underline{\pi}}(s)$ and $Q^{*,\underline{\pi}}(s,a,b) = \max_{\overline{\pi}} Q^{\overline{\pi},\underline{\pi}}(s,a,b)$. Similarly, when P2 plays the best response to P1's policy $\pi$, we define $V^{\overline{\pi},*}(s) = \min_{\underline{\pi}} V^{\overline{\pi},\underline{\pi}}(s)$ and $Q^{\overline{\pi},*}(s,a,b) = \min_{\underline{\pi}} Q^{\overline{\pi},\underline{\pi}}(s,a,b)$.

A Nash Equilibrium (NE) of the game is a pair of stochastic policies $(\overline{\pi}^*, \underline{\pi}^*)$ that are the best response to each other, which we write as $V^{\overline{\pi}^*,\underline{\pi}^*}(s) = V^{*,\underline{\pi}^*}(s) = V^{\overline{\pi}^*,*}(s)$. NE always exists for discounted two-player zero-sum Markov Games [Filar and Vrieze, 2012]. Correspondingly, let $V^*(s) := V^{\overline{\pi}^*,\underline{\pi}^*}(s)$ and $Q^*(s,a,b) := Q^{\overline{\pi}^*,\underline{\pi}^*}(s,a,b)$ denote the values of the NE. In practice, we always seek to find an $\epsilon-$optimal Nash equilibrium, which is a pair of stochastic policies $(\overline{\pi}, \underline{\pi})$ that satisfies $\mathbb{E}_{s\sim\mu_0} \max_{\overline{\pi}',\underline{\pi}'} V^{\overline{\pi}',\underline{\pi}}(s) - V^{\overline{\pi},\underline{\pi}'}(s) \leq \epsilon$. for a initial state distribution $\mu_0$.

We are interested in finding a one-sided $\epsilon-$optimal Nash equilibrium, similar to Alacaoglu et al. [2022], Zhao et al. [2022], Daskalakis et al. [2020], Zhang et al. [2019]. In particular, for the initial state distribution $\mu_0$, we seek $\underline{\pi}_{out}$ such that $\mathbb{E}_{s\sim\mu_0} \max_{\overline{\pi}} V^{\overline{\pi},\underline{\pi}_{out}}(s) - V^*(s) \leq \epsilon$.

We assume that the transition function satisfies the following assumption.

**Assumption 1.** *For each* $(s_t, a_t, b_t) \in \mathcal{S} \times \mathcal{A} \times \mathcal{A}$, *we assume that*

$$s_{t+1} = f(s_t, a_t, b_t) + \epsilon_t, \quad where \ \epsilon_t \sim \mathcal{N}(0, \sigma^2 I_d).$$

*The function* $f : \mathcal{S} \times \mathcal{A} \times \mathcal{A} \to \mathcal{S}$ *describes the general dynamics and* $\{\epsilon_t\}_{t=0}^{\infty}$ *are independent Gaussian noises. In other words,*

$$\mathbb{P}(s_{t+1}|s_t, a_t, b_t) \propto \exp\big(-\frac{\|f(s_t, a_t, b_t) - s_{t+1}\|_2^2}{2\sigma^2}\big) \tag{1}$$

### 2.2 Positive definite kernels and two decompositions

To efficiently represent the continuous state space of 2p0s-MGs which can not be handled by traditional tabular methods, it is common to embed the continuous state space into a kernel space. A widely used kernel is the positive definite (PD) kernel.

**Definition 1** ((Positive-Definite) Kernel [Mohri, 2018]). *A symmetric function* $k : \mathcal{X} \times \mathcal{X} \to \mathbb{R}$ *is said to be a positive definite kernel if for any* $\{x_1, \ldots, x_m\} \subset \mathcal{X}$, *the matrix* $\mathbf{K} = [k(x_i, x_j)]_{ij} \in \mathbb{R}^{m \times m}$ *is symmetric positive-definite.*

PD kernels admit many decompositions, such as Bochner decomposition [Devinatz, 1953], Mercer decomposition [Mercer, 1909], Canonical decomposition [Stochel, 1992] and Kolmogorov decomposition [Ghaemi et al., 2021]. Among these decompositions, Bochner decomposition [Devinatz, 1953] and Mercer decomposition [Mercer, 1909] have recently draw significant attention since they lead to efficient, low-dimensional approximations of the kernel, reducing computational complexity [Liu et al., 2021].

**Definition 2** (Bochner decomposition [Rudin, 2017] and Mercer decomposition [Mercer, 1909]). *Let* $\mathcal{X} \subset \mathbb{R}^d$ *be a compact domain,* $\mu$ *a strictly positive Borel measure on* $\mathcal{X}$, *and* $k(x, x') = G(x - x')$ *a bounded continuous shift-invariant positive definite kernel. Then,* $k(x, x')$ *admits* **Bochner decomposition**, *i.e. there exists a non-negative measure* $\omega$, *such that*

$$G(x - x') = \int_{\mathbb{R}^d} p(\omega) \exp\big(i\omega^\top(x - x')\big) \, d\omega, \tag{2}$$

*and* **Mercer decomposition**, *i.e. there exists a countable orthonormal basis* $\{e_i\}_{i=1}^{\infty}$ *of* $\mathcal{L}_2(\mu)$ *with corresponding eigenvalues* $\{\sigma_i\}_{i=1}^{\infty}$,[2] *such that*

$$k(x, x') = \sum_{i=1}^{\infty} \sigma_i e_i(x) e_i(x'), \tag{3}$$

*where the convergence is absolute and uniform for all* $(x, x') \in \mathcal{X} \times \mathcal{X}$. *Without loss of generality, we assume* $\sigma_1 \geq \sigma_2 \geq \cdots > 0$.

---

[2]Given a probability measure $\mu$ defined on $\mathcal{X} \subseteq \mathbb{R}^d$, we denote $L_2(\mu)$ as the set of functions $f : \mathcal{X} \to \mathbb{R}$ such that $\int_{\mathcal{X}} (f(x))^2 d\mu(x) < \infty$.

It can be verified that the Gaussian kernel, $k(x, x') = \exp(-\frac{\|x-x'\|^2}{2\sigma^2})$, meets the conditions in the above definition, and it admits both a Bochner decomposition and a Mercer decomposition [Ren et al., 2023]. Note that according to the Bochner/Mercer decomposition, the kernel can be represented with infinite basis. Practically, one can construct finite-dimensional approximations to the positive-definite kernel based on the Bochner/Mercer decompositions with random/Nyström features [Rahimi and Recht, 2008, Williams and Seeger, 2000], respectively. Recently, Ren et al. [2023] utilize a finite-dimensional approximation to represent the environment in stochastic nonlinear control problems and showed its superior performance, motivating our work.

# 3 Policy Optimization with Spectral Dynamic Embedding

In this section, we begin by introducing spectral dynamic embedding to represent the environment of the Markov game. This approach allows us to express the $Q$-function for any policy pair with infinite dimensional features. Subsequently, we develop finite-dimensional approximated features for computational tractability. With these features, each player conducts least square policy evaluation to estimate the $Q$-function of current policy pair based on the generated features and then improve the policy by natural policy gradient based on the estimated $Q$-function. This leads to *Spectral Dynamic Embedding Policy Optimization (SDEPO)* in Algorithm 3.

## 3.1 Spectral Dynamics Embedding

By interpreting the state transition function (1) as a Gaussian kernel, we can decompose the transition function of a Markov game using Bochner decomposition and Mercer decomposition, as detailed below.

**Lemma 1** (Spectral Dynamic Embedding). *Consider any $\alpha \in [0, 1)$. Denote $k_\alpha(x, x') = \exp\left(-\frac{(1-\alpha^2)\|x-x'\|^2}{2\sigma^2}\right)$ for any $0 \leq \alpha < 1$. We can decompose $k_\alpha(x, x')$ using Bochner decomposition and Mercer decomposition.*

*Let*

$$\psi_\omega(s, a, b) = \frac{g_\alpha(f(s, a, b))}{\alpha^d} \left[\cos\left(\frac{\omega^\top f(s, a, b)}{\sqrt{1-\alpha^2}}\right), \sin\left(\frac{\omega^\top f(s, a, b)}{\sqrt{1-\alpha^2}}\right)\right],$$

$$\chi_\omega(s') = p_\alpha(s')[\cos(\sqrt{1-\alpha^2}\omega^\top s'), \sin(\sqrt{1-\alpha^2}\omega^\top s')]^\top,$$

*where $g_\alpha(f(s, a, b)) := \exp\left(\frac{\alpha^2\|f(s,a,b)\|^2}{2(1-\alpha^2)\sigma^2}\right)$, $\omega \sim \mathcal{N}(0, \sigma^{-2}I_d)$, and $p_\alpha(s') = \frac{\alpha^d}{(2\pi\sigma^2)^{d/2}} \exp\left(-\frac{\|\alpha s'\|^2}{2\sigma^2}\right)$ is a Gaussian distribution for $s'$ with standard deviation $\frac{\sigma}{\alpha}$.*

*Using **Bochner decomposition**, we have*

$$P(s'|s, a, b) = \mathbb{E}_{\omega \sim \mathcal{N}(0, \sigma^{-2}I_d)}\left[\psi_\omega(s, a, b)^\top \chi_\omega(s')\right] := \langle \psi_\omega(s, a, b), \chi_\omega(s') \rangle_{\mathcal{N}(0, \sigma^{-2}I_d)}. \quad (4)$$

*Let $\mu$ be a strictly positive Borel measure on $\mathcal{S}$. By **Mercer's theorem**, $k_\alpha$ admits a decomposition for any $(x, x') \in \mathcal{X}$ [Fasshauer, 2011]:*

$$k_\alpha(x, x') = \sum_{i=1}^{\infty} \sigma_{\alpha,i} e_{\alpha,i}(x) e_{\alpha,i}(x'), \ \{e_{\alpha,i}\} \text{ a basis for } \mathcal{L}_2(\mu). \quad (5)$$

*We denote $k_\alpha(x, x') = \langle \tilde{e}_\alpha(x), \tilde{e}_\alpha(x') \rangle_{\ell_2}$ where $\tilde{e}_{\alpha,i}(x) = \sqrt{\sigma_{\alpha,i}} e_{\alpha,i}(x)$ for each positive integer $i$.[3]*

$$\psi_M(s, a, b) = \frac{g_\alpha(f(s, a, b))}{\alpha^d} \tilde{e}_\alpha\left(\frac{f(s, a, b)}{1-\alpha^2}\right)^\top, \chi_M(s') = p_\alpha(s')\tilde{e}_\alpha(s')^\top$$

*Then,*

$$P(s'|s, a, b) = \langle \psi_M(s, a, b), \chi_M(s') \rangle_{\ell_2}. \quad (6)$$

The tunable parameter $\alpha$ offers advantages for theoretical analysis and can also be leveraged to enhance empirical performance.

Let $\phi_\omega(\cdot) = [\psi_\omega(\cdot), r(\cdot)]$ and $\phi_M(\cdot) = [\psi_M(\cdot), r(\cdot)]$. The $\phi_\omega(\cdot)$ and $\phi_M(\cdot)$ are named as Bochner Spectral Dynamic Embedding and Mercer Spectral Dynamic Embedding, respectively. These embeddings form infinite-dimensional bases of the $Q$-function for arbitrary policy pairs, allowing for a linear representation of the $Q$-function.

---

[3]The $\ell_2$ norm is the usual norm for square-summable infinite-dimensional vectors $v$ indexed by the positive integers, such that $\|v\|_{\ell_2}^2 = \sum_{i=1}^{\infty} v_i^2$.

**Algorithm 1** Random Features Generation

---

**Data:** Transition Model $s' = f(s,a,b) + \varepsilon$ where $\varepsilon \sim \mathcal{N}(0, \sigma^2 I_d)$, Reward Function $r(s,a,b)$, Number of Random/Nyström Feature $m$

**Result:** $\phi(\cdot, \cdot, \cdot)$

1 Sample *i.i.d.* $\{\omega_i\}_{i \in [m]}$ where $\omega_i \sim \mathcal{N}(0, \sigma^{-2} I_d)$ and construct the feature

$$\psi_{\mathrm{rf}}(s,a,b) = \frac{g_\alpha(f(s,a,b))}{\alpha^d} \left[ \sin(\omega_i^\top f(s,a,b)/\sqrt{1-\alpha^2}), \cos(\omega_i^\top f(s,a,b)/\sqrt{1-\alpha^2}) \right]_{i \in [m]}, \tag{7}$$

set $\phi(s,a,b) = [\psi_{\mathrm{rf}}(s,a,b), r(s,a,b)]$.

---

**Algorithm 2** Nyström Features Generation

---

**Data:** Transition Model $s' = f(s,a,b) + \varepsilon$ where $\varepsilon \sim \mathcal{N}(0, \sigma^2 I_d)$, Reward Function $r(s,a,b)$, Number of Random/Nyström Feature $m$, Number of Nyström Samples $n_{\mathrm{Nys}} \geq m$, Nyström Sampling Distribution $\mu_{\mathrm{Nys}}$

1 Sample $n_{\mathrm{nys}}$ random samples $\{x_1, \ldots, x_{\mathrm{nys}}\}$ independently from $\mathcal{S}$, following the distribution $\mu_{\mathrm{Nys}}$.

2 Construct $n_{\mathrm{nys}}$-by-$n_{\mathrm{nys}}$ Gram matrix given by $K_{i,j}^{(n_{\mathrm{Nys}})} = k_\alpha(x_i, x_j)$.

3 Perform eigendecomposition on the Gram matrix $K^{(n_{\mathrm{nys}})} U = \Lambda U$, with $\lambda_1 \geq \cdots \geq \lambda_{n_{\mathrm{nys}}}$ denoting the corresponding eigenvalues.

4 Construct the feature

$$\psi_{\mathrm{nys}}(s,a,b) = \left[ \frac{g_\alpha(f(s,a,b))}{\alpha^d \sqrt{\lambda_i}} \sum_{\ell=1}^{n_{\mathrm{nys}}} U_{i,\ell} k_\alpha \left( x_\ell, \frac{f(s,a,b)}{1-\alpha^2} \right) \right]_{i \in [m]}, \tag{8}$$

set $\phi(s,a,b) = [\psi_{\mathrm{nys}}(s,a,b), r(s,a,b)]$.

---

**Lemma 2.** *For any policy pair, there exist weights $\{\theta_\omega^{\overline{\pi}, \underline{\pi}}\}$ (where $\omega \sim \mathcal{N}(0, \sigma^{-2} I_d)$) and $\theta_M^{\overline{\pi}, \underline{\pi}} \in \ell_2$ such that the corresponding value function satisfies*

$$Q^{\overline{\pi}, \underline{\pi}}(s,a,b) = \left\langle \phi_\omega(s,a,b), \theta_\omega^{\overline{\pi}, \underline{\pi}} \right\rangle_{\mathcal{N}(0, \sigma^{-2} I_d)} = \left\langle \phi_M(s,a,b), \theta_M^{\overline{\pi}, \underline{\pi}} \right\rangle_{\ell_2}$$

We provide the proof of this section in Appendix A.

### 3.2 Finite-dimensional truncated Embedding

Basically, a desirable feature embedding method should provide a good approximation to the underlying dynamics of MGs with a modest feature dimension. However, the dimension of both Bochner and Mercer spectral dynamic embedding is *infinite*, which is computationally intractable, motivating us to investigate the finite-dimensional embeddings.

We construct finite-dimensional truncations of Bochner and Mercer embedding using random feature [Ren et al., 2022] and Nyström feature [Williams and Seeger, 2000], respectively, to provide efficient finite linear approximations to represent the transition kernel. As show in Algorithm 1 and Algorithm 2, random feature is the Monte-Carlo approximation for the Bochner embedding and Nyström feature approximates the subspace spanned by the top eigenfunctions of the Mercer embedding via eigendecomposition of an empirical Gram matrix. A detailed derivation of the Nyström method is provided in Appendix B. We analyze the approximation error due to using the finite-dimensional basis in Appendix D. The finite-dimensional embedding is a crucial component of our algorithm, Spectral Dynamic Embedding Policy Optimization (SDEPO), which is given as Algorithm 3.

### 3.3 Policy Optimization with Finite-dimensional Embedding

After generating the finite-dimensional embedding, SDEPO is divided in two stages. It finds an approximate solution $\pi_k$ in Stage 1, which is then utilized in Stage 2 to derive an approximate solution $\overline{\pi}_k$. In each stage, there are two main components: policy evaluation and policy improvement. Least square policy evaluation is conducted for estimating the state-action value function of current policy pair upon the generated finite-dimensional truncation features $Q^{\overline{\pi}_t, \underline{\pi}_t}(s,a,b) = \phi(s,a,b)^\top w^{\overline{\pi}_t, \underline{\pi}_t}$. In the policy improvement procedure, based on the approximated value function, the natural policy gradient method is used to adjust the policy of each player iteratively in an alternating fashion.

---

**Algorithm 3** Spectral Dynamic Embedding Policy Optimization (SDEPO)

---

**Data:** Transition Model $s' = f(s,a,b) + \varepsilon$ where $\varepsilon \sim \mathcal{N}(0, \sigma^2 I_d)$, Reward Function $r(s,a,b)$, Number of Random/Nyström Feature $m$, Number of Nyström Samples $n_{\text{Nys}} \geq m$, Nyström Sampling Distribution $\mu_{\text{Nys}}$, Number of Sample $n$, Factorization Scale $\alpha$, Learning Rate $\eta$

**Result:** $\underline{\pi}_k$

5  Generate $\phi(s,a,b)$ using Algorithm 1 or Algorithm 2.

   Initialize $\overline{\theta}_0 = \underline{\theta}_0 = 0$ and $\overline{\pi}_0(\cdot|s) = \underline{\pi}_0(\cdot|s) = Unif(\mathcal{A})$ for all $s \in \mathcal{S}$.

6  **for** $k = 0, 1, \cdots, K$ **do**

     | Stage 1

7     | Initialize $\overline{\theta}_{k,0} = \overline{\theta}_k, \underline{\theta}_{k,0} = \underline{\theta}_k$ and $\overline{\pi}_{k,0}(\cdot|s) = \overline{\pi}_k(\cdot|s), \underline{\pi}_{k,0}(\cdot|s) = \underline{\pi}_k(\cdot|s)$ for all $s \in \mathcal{S}$.

8     | **for** $t = 0, 1, \cdots, T-1$ **do**

9         | Sample *i.i.d.* $\{s_i, a_i, b_i, s'_i, a'_i, b'_i\}_{i \in [n]}$ with policy pair $\overline{\pi}_{k-1}, \underline{\pi}_{k-1}$, where $s'_i = f(s_i, a_i, b_i) + \varepsilon$.

10        | Initialize $\hat{w}_{k,t,0} = 0$.

11        | **for** $l = 0, 1, \cdots, L-1$ **do**

12          | Solve

$$\hat{w}_{k,t,l+1} = \arg\min_{w} \sum_{i \in [n]} \left( \phi(s_i, a_i, b_i)^\top w - r(s_i, a_i, b_i) - \gamma \phi(s'_i, a'_i, b'_i)^\top \hat{w}_{k,t,l} \right)^2. \quad (9)$$

13        | Update $\overline{\theta}_{k,t+1} = \overline{\theta}_{k,t} + \eta \hat{w}_{k,t,L}, \underline{\theta}_{k,t+1} = \underline{\theta}_{k,t} - \eta \hat{w}_{k,t,L}$ and

$$\overline{\pi}_{k,t+1}(a|s) \propto \exp(\mathbb{E}_{b \sim \underline{\pi}_{k,t}(\cdot|s)}[\phi(s,a,b)]^\top \overline{\theta}_{k,t+1}),$$
$$\underline{\pi}_{k,t+1}(b|s) \propto \exp(\mathbb{E}_{a \sim \overline{\pi}_{k,t}(\cdot|s)}[\phi(s,a,b)]^\top \underline{\theta}_{k,t+1}). \quad (10)$$

14     | Output $\underline{\pi}_k = \frac{1}{T} \sum_{t=1}^{T} \underline{\pi}_{k,t}$.

     | Stage 2

15     | Initialize $\overline{\theta}'_{k,0} = 0$ and $\overline{\pi}'_{k,0}(\cdot|s) = Unif(\mathcal{A})$.

16     | **for** $t = 0, 1, \cdots, T-1$ **do**

17        | Sample *i.i.d.* $\{s_i, a_i, b_i, s'_i, a'_i, b'_i\}_{i \in [n]}$ with policy pair $\overline{\pi}'_{k,t}, \underline{\pi}_k$, where $s'_i = f(s_i, a_i, b_i) + \varepsilon$.

18        | Initialize $\hat{\zeta}_{k,t,0} = 0$ for all $s \in \mathcal{S}$.

19        | **for** $l = 0, 1, \cdots, L-1$ **do**

20         | Solve

$$\hat{\zeta}_{k,t,l+1} = \arg\min_{\zeta} \sum_{i \in [n]} \left( \phi(s_i, a_i, b_i)^\top \zeta - r(s_i, a_i, b_i) - \gamma \phi(s'_i, a'_i, b'_i)^\top \hat{\zeta}_{k,t,l} \right)^2. \quad (11)$$

21        | Update $\overline{\theta}'_{k,t+1} = \overline{\theta}'_{k,t} + \eta \hat{\zeta}_{k,t,L}$ and

$$\overline{\pi}'_{k,t+1}(a|s) \propto \exp(\mathbb{E}_{b \sim \underline{\pi}_k(\cdot|s)}[\phi(s,a,b)]^\top \overline{\theta}'_{k,t+1}). \quad (12)$$

22     | Output $\overline{\pi}_k = \overline{\pi}'_{k,\hat{t}}$, where $\hat{t} \in [T]$ is selected uniformly at random.

---

$\overline{\pi}_{t+1}$ is updated as

$$\overline{\pi}_{t+1}(\cdot|s) = \arg \max_{\pi(\cdot|s) \in \Delta(\mathcal{A})} \left\langle \pi(\cdot|s), \mathbb{E}_{b \sim \underline{\pi}_t(\cdot|s)}[\phi(s,\cdot,b)]^\top w^{\overline{\pi}_t, \underline{\pi}_t} \right\rangle + \frac{1}{\eta} KL \left( \pi(\cdot|s) || \overline{\pi}_t(\cdot|s) \right)$$

yielding the following closed-form solution,

$$\overline{\pi}_{t+1}(a|s) \propto \overline{\pi}_t(a|s) \exp \left( \mathbb{E}_{b \sim \underline{\pi}_t(\cdot|s)}[\phi(s,a,b)]^\top \eta w^{\overline{\pi}_t, \underline{\pi}_t} \right) = \exp \left( \mathbb{E}_{b \sim \underline{\pi}_t(\cdot|s)}[\phi(s,a,b)]^\top \overline{\theta}_{t+1} \right)$$

where $\overline{\theta}_{t+1} = \sum_{i=0}^{t} \eta w^{\overline{\pi}_i, \underline{\pi}_i}$. $\underline{\pi}$ is updated by similar update rule.

Note that in SDEPO, each player is required to have knowledge of their opponent's strategy, which is crucial for effective policy optimization and is a common requirement in numerous existing algorithms for Markov games. [Xie et al., 2020, Chen et al., 2022, Junchi Li et al., 2022, Alacaoglu et al., 2022]

# 4 Theoretical Results

The major difficulty in analyzing the convergence of SDEPO is that the policies of both players evolve at each iteration, leading to non-stationary in the environment for each player. Additionally, there exists a certain level of estimation error in the transition and Q-function for each player. As a consequence, the vanilla proof strategy of convergence of policy optimization used in majority of the literature is no longer applicable.

In this section, we provide rigorous investigation of the impact of the approximation error for policy evaluation and derive the convergence of policy optimization. We first specify the commonly used assumptions [Yu and Bertsekas, 2008, Jin et al., 2020, Agarwal et al., 2021, Abbasi-Yadkori et al., 2019, Ren et al., 2023], under which we derive our theoretical results below.

**Assumption 2** (Regularity Condition for Dynamics). *For the dynamic function $f$, there exists a constant $c_f$, such that $\|f(s,a,b)\| \le c_f$ for all $s \in \mathcal{S}, a \in \mathcal{A}, b \in \mathcal{A}$.*

**Assumption 3** (Regularity Condition for Stationary Distribution). *The stationary distribution $\nu_{\overline{\pi},\underline{\pi}}$ for all policy pair $(\overline{\pi}, \underline{\pi})$ has full support, and satisfies the following conditions with $\Upsilon_1, \Upsilon_2 > 0$:*

$$\lambda_{\min} \left( \mathbb{E}_{\nu_{\overline{\pi},\underline{\pi}}} \left[ \phi(s,a,b)\phi(s,a,b)^\top \right] \right) \ge \Upsilon_1,$$

$$\lambda_{\min} \left( \mathbb{E}_{\nu_{\overline{\pi},\underline{\pi}}} \left[ \phi(s,a,b) \left( \phi(s,a,b) - \gamma \mathbb{E}_{\nu_{\overline{\pi},\underline{\pi}}} \phi(s',a',b') \right)^\top \right] \right) \ge \Upsilon_2.$$

**Assumption 4** (Regularity Condition for Feature). *The features $\{\phi_\omega(s,a,b)\}_{(s,a,b)\in\mathcal{S}\times\mathcal{A}\times\mathcal{A}}$ and $\{\phi_\omega(s,a,b)\}_{(s,a,b)\in\mathcal{S}\times\mathcal{A}\times\mathcal{A}}$ are linearly independent.*

First, we have the following bound on the error for policy evaluation. For convenience, we focus solely on the error for least square policy evaluation in Stage 1. A similar analysis can be conducted for policy evaluation error in Stage 2. We decompose the error for policy evaluation into two parts, one is the approximation error due to the finite number of features, and one is the statistical error due to the finite number of samples. Combine the approximation error and the statistical error, we have the following bound on the error for policy evaluation. Detailed proof can be found in Appendix D.

**Theorem 1.** *Let $L = \Theta(\log n)$. Denote $\tilde{g}_\alpha := \sup_{s,a,b} \frac{g_\alpha(f(s,a,b))}{\alpha^d}$ and $\hat{Q}^{\overline{\pi},\underline{\pi}}_{\Phi,L} = \phi^\top \hat{w}_L$. With probability at least $1 - \delta$, we have that for the random features,*

$$\left\| Q^{\overline{\pi},\underline{\pi}} - \hat{Q}^{\overline{\pi},\underline{\pi}}_{\Phi_{\mathrm{rf}},L} \right\|_{\nu_{\overline{\pi},\underline{\pi}}} = \tilde{O} \left( \frac{\tilde{g}_\alpha}{(1-\gamma)^2\sqrt{m}} + \frac{\tilde{g}_\alpha^6 m^3}{(1-\gamma)\Upsilon_1^2\Upsilon_2\sqrt{n}} \right), \tag{13}$$

*and for the Nyström features,*

$$\left\| Q^{\overline{\pi},\underline{\pi}} - \hat{Q}^{\overline{\pi},\underline{\pi}}_{\Phi_{\mathrm{nys}},L} \right\|_{\nu_{\overline{\pi},\underline{\pi}}} = \tilde{O} \left( \frac{\tilde{g}_\alpha}{(1-\gamma)^2 n_{\mathrm{nys}}} + \frac{\tilde{g}_\alpha^6 m^3}{(1-\gamma)\Upsilon_1^2\Upsilon_2\sqrt{n}} \right). \tag{14}$$

As shown in Theorem D.1, Nyström method improves the approximation error from $O(m^{-1})$ to $O(n_{nys}^{-1})$ with a mild assumption to make in the kernel literature (cf. Theorem 2 in [Belkin, 2018]).

Then, we provide the error analysis for policy optimization. We remark that in the following proof we assume the action space $|\mathcal{A}|$ is finite for simplicity. The following assumption assumes that the selection probability of each action is positive under each state, which is a commonly used assumption in the analysis of policy gradient type methods [Alacaoglu et al., 2022, Lan, 2023].[4]

**Assumption 5.** *There exists a constant $\underline{c}$ such that, for any policy iterate pair $\overline{\pi}_k, \underline{\pi}_k$, for any state action tuple $s, a, b$, it holds that $\overline{\pi}_k(a|s) \ge \underline{c} > 0, \underline{\pi}_k(b|s) \ge \underline{c} > 0$.*

We now present the result for policy optimization. We use Perolat et al. [2015]'s error propagation framework, which needs the results in each stage. The detailed proof of it can be found in Appendix E. First, we analyze the error in the Stage 1 of Algorithm 3.

**Lemma 3.** *Denote $Q_{k-1} = Q^{\overline{\pi}_{k-1},\underline{\pi}_{k-1}}$, $\overline{\pi}^s Q^s \underline{\pi}^s = \mathbb{E}_{a\sim\overline{\pi}(\cdot|s),b\sim\underline{\pi}(\cdot|s)}[Q(s,a,b)]$. Let Assumption 5 hold. In Stage 1 of Algorithm 3,*

$$\mathbb{E} \left[ \max_{\overline{\pi}} \overline{\pi} Q_{k-1}\underline{\pi}_k - \min_{\underline{\pi}} \overline{\pi}_k Q_{k-1}\underline{\pi} \right]$$

$$\le \frac{1}{\eta T} \log \frac{1}{\underline{c}} + 2\|\hat{Q}^{\overline{\pi}_{k-1},\underline{\pi}_{k-1}} - Q^{\overline{\pi}_{k-1},\underline{\pi}_{k-1}}\|_{\nu_{\overline{\pi}_{k-1},\underline{\pi}_{k-1}}} + \frac{\eta}{2(1-\gamma)^2}.$$

---

[4]This assumption can be removed if we use a $\beta-$greedy mixed version of $\overline{\pi}_k, \underline{\pi}_k$. As the $\beta-$greedy mixed policies are shifted with the original ones, we provide the convergence analyses of SDEPO with $\beta-$greedy exploration in Appendix E.1.

This result shows how the error in the approximation of the $Q$-function and the learning rate $\eta$ impact the optimization performance in Stage 1. Next, we turn to Stage 2 and provide an upper bound on the one-sided error.

**Lemma 4.** *Let Assumption 5 hold and $\mu_0$ be a state distribution. In Stage 2 of Algorithm 3,*

$$\frac{1}{T}\sum_{t=1}^{T}\mathbb{E}_{s\sim\mu_0}\left[V^{\overline{\pi}_k^*,\underline{\pi}_k}(s) - V^{\overline{\pi}_{k,t}',\underline{\pi}_k}(s)\right]$$

$$\leq\frac{1}{1-\gamma}\left(\frac{\log\frac{1}{\underline{c}}}{\eta T} + \frac{\eta}{(1-\gamma)^2} + \eta\|\hat{Q}^{\overline{\pi}_{k,t}',\underline{\pi}_k} - Q^{\overline{\pi}_{k,t}',\underline{\pi}_k}\|_{\nu_{\overline{\pi}_{k,t}',\underline{\pi}_k}}\right).$$

Building on the results from Stage 1 and Stage 2, along with the policy evaluation error, and using the error propagation framework from Perolat et al. [2015], we can now derive the overall convergence result for the policy optimization process. Specifically, we obtain the following proposition, which quantifies the iteration complexity required to reach a near-optimal solution:

**Proposition 1.** *Let* $L = \Theta(\log n)$ *and* $T = \sqrt{\frac{\log 1/c}{\log n}}\left(\frac{1}{(1-\gamma)^2} + \frac{\tilde{g}_\alpha}{(1-\gamma)^2\sqrt{m}} + \frac{\tilde{g}_\alpha^6 m^3}{(1-\gamma)\Upsilon_1^2\Upsilon_2\sqrt{n}}\right)$ *for*

*random features and* $T = \sqrt{\frac{\log 1/c}{\log n}}\left(\frac{1}{(1-\gamma)^2} + \frac{\tilde{g}_\alpha}{(1-\gamma)^2 n_{\text{nys}}} + \frac{\tilde{g}_\alpha^6 m^3}{(1-\gamma)\Upsilon_1^2\Upsilon_2\sqrt{n}}\right)$ *for Nyström features.*

*Iteration complexity to get* $E_{s\sim\mu_0}\left[\max_{\overline{\pi}}V^{\overline{\pi},\underline{\pi}_k}(s) - V^*(s)\right] \leq \epsilon$ *is* $\widetilde{O}(\frac{1}{(1-\gamma)^3\epsilon})$.

Proposition 1 shows the iteration complexity to one-sided NE and it can be directly extended to establish a two-sided NE by applying the algorithm with the roles switched [Zhao et al., 2022].

# 5 The Practical variant of SDEPO

Many practical 2p0s-MGs not only have continuous state spaces but also continuous action spaces, such as such as real-time strategy games [Vinyals et al., 2019, Berner et al., 2019] and robust policy optimization [Pinto et al., 2017]. Note that the natural policy gradient update (31) and (12) in SDEPO involve the expectation w.r.t. the action, which is generally intractable for continuous action space. Hence, we propose a practical variant of SDEPO, named SDEPO-NN, to deal with continuous (or large-scale discrete ) action space. SDEPO-NN utilizes neural networks in policy $\pi$ and the state-action value function approximation $Q$, detailed in Algorithm F.

Based on the spectral dynamic embedding $\phi$, we parameterize the $\overline{Q}$ function of player one as $\overline{Q}_{\overline{\theta}}(s,a,b) = r(s,a,b) + \phi(s,a,b)^\top\overline{\theta}$ and parameterize the $\underline{Q}$ function of player two as $\underline{Q}_{\underline{\theta}}(s,a,b) = r(s,a,b) + \phi(s,a,b)^\top\underline{\theta}$, and train $\overline{Q}_{\overline{\theta}}$ and $\underline{Q}_{\underline{\theta}}$ by minimizing the soft Bellman residual.

We restrict the players' policies $\overline{\pi}_{\overline{\psi}}$ and $\underline{\pi}_{\underline{\psi}}$ to Gaussians with the reparametrization trick, i.e., $a_t = f_{\overline{\psi}}(\epsilon_t; s_t)$ and $b_t = f_{\underline{\psi}}(\epsilon_t'; s_t)$ where $\epsilon_t$ and $\epsilon_t'$ are input noise vectors, sampled from a Gaussian. The policy parameters can be learned by minimizing

$$J(\overline{\psi}) = \mathbb{E}_{s_t\sim\mathcal{D},\epsilon_t,\epsilon_t'\sim\mathcal{N}}\left[\log\overline{\pi}_{\overline{\psi}}(f_{\overline{\psi}}(\epsilon_t; s_t)|s_t) - \overline{Q}_{\overline{\theta}}(s_t, \overline{\pi}_{\overline{\psi}}(f_{\overline{\psi}}(\epsilon_t; s_t)|s_t), \underline{\pi}_{\underline{\psi}}(f_{\underline{\psi}}(\epsilon_t'; s_t)|s_t))\right],$$

and

$$J(\underline{\psi}) = \mathbb{E}_{s_t\sim\mathcal{D},\epsilon_t,\epsilon_t'\sim\mathcal{N}}\left[\log\underline{\pi}_{\underline{\psi}}(f_{\underline{\psi}}(\epsilon_t'; s_t)|s_t) - \underline{Q}_{\underline{\theta}}(s_t, \overline{\pi}_{\overline{\psi}}(f_{\overline{\psi}}(\epsilon_t; s_t)|s_t), \underline{\pi}_{\underline{\psi}}(f_{\underline{\psi}}(\epsilon_t'; s_t)|s_t))\right],$$

where $\mathcal{D}$ is the replay buffer, and $\overline{\pi}_{\overline{\psi}}$ and $\underline{\pi}_{\underline{\psi}}$ are defined implicitly in terms of $f_{\overline{\psi}}$ and $f_{\underline{\psi}}$, respectively.

Note that tabular methods [Alacaoglu et al., 2022, Bai and Jin, 2020, Daskalakis et al., 2020, Wei et al., 2021, Zhao et al., 2022] can discretize the state/action spaces to handle applications with continuous state/action spaces. However, directly applying such methods often incurs the curse of dimension and requires an excessive amount of computation resources even for a small size problem. On the other hand, although there exist theoretical-guaranteed methods for 2p0s-MGs with continuous state and action space, they all involve a computational intractable subroutine, i.e., Qiu et al. [2021], Junchi Li et al. [2022] need to solve a difficult 'find_ne'/'find_cce' subroutine, and Jin et al. [2022], Huang et al. [2021] have to tackle a comprehensive constrained optimization problem.

# 6 Numerical verification

In this section, we present two experiments to evaluate our methods. The first experiment focuses on a simple zero-sum Markov game featuring a continuous state space and a finite action space, aiming

to validate the convergence of SDEPO. The second experiment adapts a multi-agent scenario inspired by the simple push [Lowe et al., 2017], where both the state and action spaces are continuous, to assess the effectiveness of SDEPO-NN.

In the first experiment, we designed a simple zero-sum Markov game with a continuous state and finite action space ($\mathcal{S} = \mathbb{R}$, $|\mathcal{A}| = 5$). The state space is partitioned into 42 distinct intervals: one interval for $(-\infty, -10)$, 40 intervals evenly spaced by 0.5 units in the range $[-10, 10)$, and one interval for $(10, \infty)$. In the $i$-th interval, the transition dynamics are defined by $P(s, a, b) = f(s, a, b) + \epsilon$, where $\epsilon \sim \mathcal{N}(0, 1)$, and $f(s, a, b) = \epsilon_{i,a,b}$, with $\epsilon_{i,a,b} \sim \text{Unif}(-10.5, 10.5)$. The reward function is $r(s, a, b) = \epsilon'_{i,a,b}$, where $\epsilon'_{i,a,b} \sim \text{Unif}(-1, 1)$. The initial state distribution is assumed to be uniform over $[-10.5, 10.5]$.

We ran SDEPO for 120 iterations, and measured the convergence of $\underline{\pi}$ by metrics in Proposition 1. As shown in Figure 1, SDEPO with random features and Nyström features both converge after 60 iterations. We discretized the state space of this environment and compared it with OFTRL [Zhang et al., 2022], a tabular method where the environment is known. We adopted the parameter settings recommended in [Zhang et al., 2022] and adjusted the environment to a 100-horizon setting. As shown in Figure 1, our method demonstrated superior convergence in this environment. This likely stems from the fact that OFTRL operates on the discretized state space, whereas our method computes on the original state space.

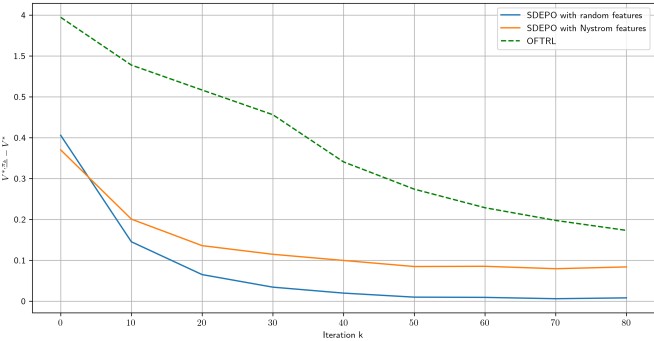

Figure 1: Performance illustration of SDEPO and OFTRL for solving the random generated Markov game.

Next, we conduct experiments on an adapted version of simple push [Lowe et al., 2017], wherein both the state and action spaces are continuous. This problem consists of two agents and one landmark. Each agent receives a reward for proximity to the landmark while ensuring the other agent remains distant. Thus, agents must learn to stay close to the landmark and simultaneously push the other agent away. At each time step, a noise $\epsilon \sim \mathcal{N}(0, \sigma^2 I_d)$ is added to the state.

We implemented SDEPO-NN with random features (SDEPO-NN$_{rf}$) and Nyström features (SDEPO-NN$_{nys}$), comparing them against methods where Q functions do not utilize spectral dynamical embedding (NPG-NN). Table 2 shows the results of winning rate after training by 20000 iterations with varying noise levels. It is evident that SDEPO-NN$_{rf}$ and SDEPO-NN$_{nys}$ largely outperforms NPG-NN, which shows the effectiveness of the spectral dynamical embedding.

| Winning rate | NPG-NN | SDEPO-NN$_{rf}$ | SDEPO-NN$_{nys}$ |
|---|---|---|---|
| NPG-NN | 49.69%/47.7% | 5.23%/0.55% | 5.16%/0.59% |
| SDEPO-NN$_{rf}$ | 95.13%/99.37% | 50.35%/49.95% | 49.74%/49.94% |
| SDEPO-NN$_{nys}$ | 95.38%/99.38% | 49.69%/49.95% | 50%/49.84% |

Table 2: Comparison of winning rates between NPG-NN, SDEPO-NN$_{rf}$, and SDEPO-NN$_{nys}$ in Simple Push with $\sigma = 0.1/0.01$. The results before and after / correspond to $\sigma = 0.1$ and $0.01$, respectively.

## 7  Conclusion

In this paper, we propose a provably efficient natural policy gradient algorithm for two-player zero-sum stochastic Markov games with continuous state. We analyze the approximation error and convergence of the algorithm. To deal with continuous action spaces, a practical variant is provided and demonstrates superior performance. A possible direction is to extend our methods to independent learning setting.

## Acknowledgments

This work was supported in part by National Natural Science Foundation of China under Grant 62206248 and National Natural Science Foundation of China under Grant 62402430.

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

# A  Derivation of Spectral Dynamic Embedding

In this section, we derivate spectral dynamic embedding and then prove the Q-function for arbitrary policy pair can be linearly represented by the feature functions $\{\phi_\omega(s, a, b)\}$ with $\omega \sim \mathcal{N}\left(0, \sigma^2 I_d\right)$ and $\phi_M(s, a, b) \in \ell_2$.

**Lemma A.1** (i.e. Lemma 1). *Consider any $\alpha \in [0, 1)$. Denote $k_\alpha(x, x') = \exp\left(-\frac{(1-\alpha^2)\|x-x'\|^2}{2\sigma^2}\right)$ for any $0 \le \alpha < 1$. We can decompose $k_\alpha(x, x')$ using Bochner decomposition and Mercer decomposition. Let*

$$\psi_\omega(s, a, b) = \frac{g_\alpha(f(s, a, b))}{\alpha^d}\left[\cos\left(\frac{\omega^\top f(s, a, b)}{\sqrt{1-\alpha^2}}\right), \sin\left(\frac{\omega^\top f(s, a, b)}{\sqrt{1-\alpha^2}}\right)\right],$$

$$\chi_\omega(s') = p_\alpha(s')[\cos(\sqrt{1-\alpha^2}\omega^\top s'), \sin(\sqrt{1-\alpha^2}\omega^\top s')]^\top,$$

*where $g_\alpha(f(s, a, b)) := \exp\left(\frac{\alpha^2\|f(s,a,b)\|^2}{2(1-\alpha^2)\sigma^2}\right)$, $\omega \sim \mathcal{N}(0, \sigma^{-2}I_d)$, and $p_\alpha(s') = \frac{\alpha^d}{(2\pi\sigma^2)^{d/2}}\exp\left(-\frac{\|\alpha s'\|^2}{2\sigma^2}\right)$ is a Gaussian distribution for $s'$ with standard deviation $\frac{\sigma}{\alpha}$. Using Bochner decomposition, we have*

$$P(s'|s, a, b) = \mathbb{E}_{\omega \sim \mathcal{N}(0, \sigma^{-2}I_d)}\left[\psi_\omega(s, a, b)^\top \chi_\omega(s')\right] := \langle\psi_\omega(s, a, b), \chi_\omega(s')\rangle_{\mathcal{N}(0, \sigma^{-2}I_d)}, \quad (15)$$

*By Mercer's theorem, $k_\alpha$ admits a decomposition for any $(x, x') \in \mathcal{X}$:*

$$k_\alpha(x, x') = \sum_{i=1}^{\infty}\sigma_{\alpha,i}e_{\alpha,i}(x)e_{\alpha,i}(x'), \ \{e_{\alpha,i}\} \text{ a basis for } \mathcal{L}_2(\mu). \quad (16)$$

*We denote $k_\alpha(x, x') = \langle\tilde{e_\alpha}(x), \tilde{e_\alpha}(x')\rangle_{\ell_2}$ where $\tilde{e}_{\alpha,i}(x) = \sqrt{\sigma_{\alpha,i}}e_{\alpha,i}(x)$ for each positive integer $i$.[5]*

$$\psi_M(s, a, b) = \frac{g_\alpha(f(s, a, b))}{\alpha^d}\tilde{e}_\alpha\left(\frac{f(s, a, b)}{1-\alpha^2}\right)^\top, \chi_M(s') = p_\alpha(s')\tilde{e}_\alpha(s')^\top$$

*Then,*

$$P(s'|s, a, b) = \langle\psi_M(s, a, b), \chi_M(s')\rangle_{\ell_2}. \quad (17)$$

*Proof.* For Bochner decomposition, we first notice that $\forall \alpha \in (0, 1)$, we have

$$P(s'|s, a, b) \propto \exp\left(-\frac{\|s' - f(s, a, b)\|^2}{2\sigma^2}\right)$$

$$= \exp\left(-\frac{\|\alpha s'\|^2}{2\sigma^2}\right)\exp\left(-\frac{\|(1-\alpha^2)s' - f(s, a, b)\|^2}{2\sigma^2(1-\alpha^2)}\right)\exp\left(\frac{\alpha^2\|f(s, a, b)\|^2}{2(1-\alpha^2)\sigma^2}\right). \quad (18)$$

The factorization of the transition $P(s'|s, a, b)$ in (15) can thus be derived from the property of the Gaussian kernel by applying Bochner decomposition in Lemma 2 to the second term in (18).

For the Mercer decomposition, the proof is analogous, where here we apply Mercer's theorem to decompose the middle term in terms of the kernel $k_\alpha$

$$k_\alpha\left(s', \frac{f(s, a, b)}{1-\alpha^2}\right) = \exp\left(-\frac{(1-\alpha^2)\left\|s' - \frac{f(s,a,b)}{1-\alpha^2}\right\|^2}{2\sigma^2}\right) = \exp\left(-\frac{\|(1-\alpha^2)s' - f(s, a, b)\|^2}{2\sigma^2(1-\alpha^2)}\right).$$

$\square$

---

[5] The $\ell_2$ norm is the usual norm for square-summable infinite-dimensional vectors $v$ indexed by the positive integers, such that $\|v\|_{\ell_2}^2 = \sum_{i=1}^{\infty}v_i^2$.

Let $\phi_\omega(\cdot) = [\psi_\omega(\cdot), r(\cdot)]$ and $\phi_M(\cdot) = [\psi_M(\cdot), r(\cdot)]$. Here we prove the Q-function for arbitrary policy pair can be linearly represented by the feature functions $\{\phi_\omega(s, a, b)\}$ with $\omega \sim \mathcal{N}\left(0, \sigma^2 I_d\right)$ and $\phi_M(s, a, b) \in \ell_2$.

**Lemma A.2** (i.e. Lemma 2). *For any policy, there exist weights $\{\theta_\omega^{\overline{\pi}, \underline{\pi}}\}$ (where $\omega \sim \mathcal{N}(0, \sigma^{-2} I_d)$) and $\theta_M^{\overline{\pi}, \underline{\pi}} \in \ell_2$ such that the corresponding value function satisfies*

$$Q^{\overline{\pi}, \underline{\pi}}(s, a, b) = \left\langle \phi_\omega(s, a, b), \theta_\omega^{\overline{\pi}, \underline{\pi}} \right\rangle_{\mathcal{N}(0, \sigma^{-2} I_d)} = \left\langle \phi_M(s, a, b), \theta_M^{\overline{\pi}, \underline{\pi}} \right\rangle_{\ell_2}$$

*Proof.* Denote $\mu_\omega(\cdot) = [\chi_\omega(\cdot), 0]^\top$, $\mu_M(\cdot) = [1, \chi_M(\cdot)]^\top$, $\theta_r = [0, 0, 1]^\top$, $\theta_{r,M} = [1, 0, 0, \dots]^\top \in \ell_2$. Our claim can be verified easily by applying the decompositions to Bellman recursion:

$$Q^{\overline{\pi}, \underline{\pi}}(s, a, b) = r(s, a, b) + \gamma \mathbb{E}_P\left[V^{\overline{\pi}, \underline{\pi}}(s')\right] \tag{19}$$

$$= \left\langle \phi_\omega(s, a, b), \underbrace{\theta_r + \gamma \int_{\mathcal{S}} \mu_\omega(s') V^{\overline{\pi}, \underline{\pi}}(s') ds'}_{\theta_\omega^{\overline{\pi}, \underline{\pi}}} \right\rangle_{\mathcal{N}(0, \sigma^{-2} I_d)}$$

$$= \left\langle \phi_M(s, a, b), \underbrace{\theta_{r,M} + \gamma \int_{\mathcal{S}} \mu_M(s') V^{\overline{\pi}, \underline{\pi}}(s') ds'}_{\theta_M^{\overline{\pi}, \underline{\pi}}} \right\rangle_{\ell_2},$$

where the second equation comes from the Bochner decomposition and the third equation comes from the Mercer decomposition.

$\square$

## B   Derivation of Nyström method

Consider a bounded and continuous positive definite kernel $k(x, y)$ defined on a compact space $\mathcal{S}$, along with any probability measure $\mu$ on $\mathcal{S}$, e.g. the $\mu_{\text{nys}}$ considered in Algorithm 3. Mercer's theorem guarantees the existence of eigenvalues $\{\sigma_j\}_{j=1}^\infty$ and orthonormal eigenvectors $\{e_j\}_{j=1}^\infty \subset \mathcal{L}_2(\mu)$ such that for any $j \in \mathcal{N}$ and any $x \in \mathcal{S}$, the following holds:

$$\int_{\mathcal{S}} k(x, y) e_j(y) d\mu(y) = \sigma_j e_j(x). \tag{20}$$

The Nyström method provides an approximation of the Mercer eigendecomposition of $k(x, y) = \sum_{j=1}^\infty \sigma_j e_j(x) e_j(y)$ in the form

$$\hat{k}_m^n(x, y) = \sum_{i=1}^m \hat{\sigma}_i \hat{e}_i(x) \hat{e}_i(y)$$

for some positive integer $m$. Here, the pairs where the $(\hat{\sigma}_i, \hat{e}_i)$ are determined through a numerical approximation of the eigenfunction problem outlined (20). We will now describe the procedure in detail.

Recall that $n_{\text{nys}}$ refers to the number of samples used to construct the Nyström features. For simplicity, in the sections of the Appendix that address Nyström approximation results, we denote $n := n_{\text{nys}}$, unless otherwise noted. It is important not to confuse this $n$ with the $n$ used to represent the number of samples for statistical learning of the $Q$-function. Suppose we draw $n$ independent samples $X^n = \{x_s\}_{s=1}^n$ from $\mathcal{S}$ according to the distribution $\mu$. For any eigenfunction $e_j$, the eigenfunction problem from (20) can be numerically approximated by:

$$\frac{1}{n} \sum_{s=1}^n k(x, x_s) e_j(x_s) \approx \sigma_j e_j(x), \tag{21a}$$

$$\frac{1}{n} \sum_{s=1}^n (e_j(x_s))^2 \approx 1. \tag{21b}$$

Let $K^{(n)} \in \mathbb{R}^{n \times n}$ denote the Gram matrix where $(K^n)_{rs} = k(x_r, x_s)$, and let the eigendecomposition of $K^{(n)}$ be $K^{(n)} U = \Lambda U$ with $U$ orthogonal and $\Lambda$ diagonal. To satisfy (21a), the condition must hold for all $x \in X^n$. Specifically, for any $r \in [n]$, we should expect

$$\frac{1}{n} \sum_{s=1}^{n} k(x_r, x_s) e_j(x_s) \approx \sigma_j e_j(x_r) \tag{22}$$

for any eigenfunction $e_j(\cdot)$ and its corresponding eigenvalue $\sigma_j$ of $k(\cdot, \cdot)$. For any eigenvector $u_i$ of the Gram matrix $K^{(n)}$, we have the relation

$$\frac{1}{n} \sum_{s=1}^{n} k(x_r, x_s)(u_i)_s = \frac{\lambda_i}{n}(u_i)_r.$$

It is therefore natural to extend $u_i \in \mathbb{R}^n$ to an eigenfunction $\tilde{e}_i$ by setting $\tilde{e}_i(x_s) = (u_i)_s$ for any $x_s \in X^n$, and for other $x$, define

$$\tilde{e}_i(x) := \frac{1}{\lambda_i} \sum_{s=1}^{n} k(x, x_s)(u_i)_s,$$

with associated eigenvalue $\frac{\lambda_i}{n}$. To satisfy the orthonormality condition from (21b), since

$$\frac{1}{n} \sum_{s=1}^{n} \tilde{e}_i(x_s)^2 = \frac{1}{n} \sum_{s=1}^{n} (u_i)_s^2 = \frac{1}{n},$$

we scale

$$\hat{e}_i(x) := \sqrt{n} \tilde{e}_i(x) = \frac{\sqrt{n}}{\lambda_i} \sum_{s=1}^{n} k(x, x_s)(u_i)_s.$$

Using these scaled eigenfunctions, we define the Nyström approximation $\hat{k}^n$ of the original kernel $k$ as

$$\hat{k}^n(x, y) = \sum_{i=1}^{n} \frac{\lambda_i}{n} \hat{e}_i(x) \hat{e}_i(y) = k(x, X^n) \left( \sum_{i=1}^{n} \frac{1}{\lambda_i} u_i u_i^\top \right) k(X^n, x), \tag{23}$$

where $k(x, X^n)$ is a row vector with components $k(x, x_s)$, and $k(X^n, y)$ is a column vector with components $k(x_s, y)$. Based on (23), for any $m \leq n$, we then define the rank-$m$ Nyström approximation as

$$\hat{k}_m^n(x, y) = k(x, X^n) \left( \sum_{i=1}^{m} \frac{1}{\lambda_i} u_i u_i^\top \right) k(X^n, x), \tag{24}$$

which is the rank-$m$ Nyström kernel approximation used in the paper. Since it can be written as $\hat{k}_{,m}^n(x, y) = \varphi_{\text{nys}}(x)^\top \varphi_{\text{nys}}(y)$, where

$$(\varphi_{\text{nys}})_i(\cdot) = \frac{1}{\sqrt{\lambda_i}} u_i^\top k(X^n, \cdot), \quad \forall i \in [m], \tag{25}$$

we can view $\varphi_{\text{nys}}(\cdot) \in \mathbb{R}^m$ as the rank-$m$ Nyström features corresponding to the Nyström approximation.

## C  The performance difference lemma

**Lemma C.3** (Performance difference lemma.  See [Alacaoglu et al., 2022]). *For any policies $\pi_1, \pi_2, \underline{\pi}$ and any state $s_0$,*

$$V^{\pi_1, \underline{\pi}}(s_0) - V^{\pi_2, \underline{\pi}}(s_0) = \frac{1}{1 - \gamma} \mathbb{E}_{s' \sim d_{s_0}^{\pi_1, \underline{\pi}}} \langle \mathbb{E}_{b \sim \underline{\pi}(\cdot|s)} Q^{\pi_2, \underline{\pi}}(s, \cdot, b), \pi_1(\cdot|s) - \pi_2(\cdot|s) \rangle$$

*where $d_{s_0}^{\pi_1, \pi_2}(s) = (1 - \gamma) \sum_{t=0}^{\infty} \gamma^t \mathbb{P}^{\pi_1, \pi_2}(s_t = s|s_0)$, where $\mathbb{P}^{\pi_1, \pi_2}(s_t = s|s_0)$ denote the probability that $s_t = s$ after starting at $s_0$ and following the policies $\pi_1$ and $\pi_2$.*

# D  Error analysis for Policy Evaluation

We provide a brief outline of our overall proof strategy for policy evaluation. For convenience, we focus solely on the error for least square policy evaluation in Stage 1. A similar analysis can be conducted for policy evaluation error in Stage 2.

We decompose the error into two parts, one is the approximation error due to the limitation of our basis (*i.e.*, finite $m$ in Line 1 of Algorithm 1 and Line 4 of Algorithm 2), and one is the statistical error due to the finite number of samples we use (*i.e.*, finite $n$ in Line 27 of Algorithm 3). For notational simplicity, we omit $\overline{\pi}, \underline{\pi}$ and use $\nu$ to denote the stationary distribution corresponding to $\overline{\pi}, \underline{\pi}$ in this section. For the ease of presentation, we omit the polynomial dependency on $c_f$ and focus on the dependency of other terms of interest.

## D.1  Approximation Error

We start by deriving a bound on the approximation error when representing the $Q$-function for a given policy pair using an imperfect, finite-dimensional feature set. In such cases, the best possible approximation, denoted as $\tilde{Q}$, is the solution to a projected Bellman equation (cf. Yu and Bertsekas [2008]), which is defined as follows: given any (possibly finite-dimensional) feature map $\Phi := \{\phi(s, a, b)\}_{(s,a,b)\in\mathcal{S}\times\mathcal{A}\times\mathcal{A}}$, the approximation $\tilde{Q}\Phi^{\overline{\pi},\underline{\pi}}$ is defined by:

$$\tilde{Q}\Phi^{\overline{\pi},\underline{\pi}} = \Pi\nu, \Phi(r + P^{\overline{\pi},\underline{\pi}}\tilde{Q}\Phi^{\overline{\pi},\underline{\pi}}), \tag{26}$$

where $\nu$ represents the stationary distribution under $\overline{\pi}, \underline{\pi}$, and the operator $P^{\overline{\pi},\underline{\pi}}$ is given by

$$(P^{\overline{\pi},\underline{\pi}}f)(s, a, b) = \mathbb{E}_{(s',a',b')\sim P(s,a,b)\times\overline{\pi}\times\underline{\pi}}f(s', a', b'). \tag{27}$$

and $\Pi_{\nu,\Phi}$ is the projection operator as defined as

$$\Pi_{\nu,\Phi}Q = \underset{f\in\text{span}(\Phi)}{\arg\min} \ \mathbb{E}_{\nu} \left(Q(s, a, b) - f(s, a, b)\right)^2. \tag{28}$$

Our focus on $\tilde{Q}_\Phi^\pi$ is motivated by the fact that the least-squares policy evaluation step in Algorithm 3 (see equation (11)) recovers $\tilde{Q}_\Phi^\pi$ if the number of samples, $n$, goes to infinity. The effect of a finite sample size $n$ on the statistical error will be discussed later.

Next, we present our bound on the approximation error for the random and Nyström features. We begin with the random feature approach, which . To do so, we first need the following technical result, which relies on the following technical lemma, adapted from Lemma 1 in Rahimi and Recht [2008].

**Lemma D.4** (cf. Lemma 1 from Rahimi and Recht [2008]). *Let $p$ be a distribution on a space $\Omega$, and consider a mapping $\phi(x; \omega) \in \mathbb{R}^\ell$. Suppose*

$$f^*(x) = \int_\Omega p(\omega)\beta(\omega)^\top \phi(x; \omega)d\omega,$$

*for some vector $\beta(\omega) \in \mathbb{R}^\ell$ where $\sup_{x,\omega} \left|\beta(\omega)^\top\phi(x;\omega)\right| \leq C$ for some $C > 0$. Consider $\{\omega_i\}_{i=1}^k$ drawn iid from $p$, and denote the sample average of $f^*$ as $\hat{f}(x) = \frac{1}{K}\sum_{k=1}^K \beta(\omega_k)^\top\phi(x;\omega_k)$. Then, for any $\delta > 0$, with probability at least $1 - \delta$ over the random draws of $\{\omega_i\}_{i=1}^k$,*

$$\sqrt{\int_{\mathcal{X}} \left(\hat{f}(x) - f^*(x)\right)^2 d\mu(x)} \leq \frac{C}{\sqrt{K}}\left(1 + \sqrt{2\log\frac{1}{\delta}}\right).$$

Based on Lemma D.4, we derive the approximation error with random features.

**Proposition D.1** ($Q$-Approximation error with random features). *We define the feature map $\Phi_{\mathrm{rf}}$ for random features as follows*

$$\Phi_{\mathrm{rf}} = \{[\psi_{\mathrm{rf}}(s, a, b), r(s, a, b)]\}_{(s,a,b)\in\mathcal{S}\times\mathcal{A}\times\mathcal{A}}, \tag{29}$$

*where $\psi_{\mathrm{rf}}(s, a, b)$ is defined in (7) in Algorithm 3. Then, for any $\delta > 0$, with probability at least $1 - \delta$, we have that*

$$\left\|Q^{\overline{\pi},\underline{\pi}} - \tilde{Q}_{\Phi_{\mathrm{rf}}}^{\overline{\pi},\underline{\pi}}\right\|_\nu = \tilde{O}\left(\frac{\gamma\tilde{g}_\alpha}{(1-\gamma)^2\sqrt{m}}\right), \tag{30}$$

*where $\|\cdot\|_\nu$ is the $L_2$ norm defined as $\|f\|_\nu = \int f^2 d\nu$, and $\tilde{Q}_{\Phi_{\mathrm{rf}}}^{\overline{\pi},\underline{\pi}}$ is defined in (26).*

*Proof.* By leveraging the contraction property and the results from Yu and Bertsekas [2008], we establish the following bound:

$$\left\| Q^{\overline{\pi},\underline{\pi}} - \tilde{Q}^{\overline{\pi},\underline{\pi}}_{\Phi_{\mathrm{rf}}} \right\|_{\nu} \leq \frac{1}{1-\gamma} \left\| Q^{\overline{\pi},\underline{\pi}} - \Pi_{\nu,\Phi_{\mathrm{rf}}} Q^{\overline{\pi},\underline{\pi}} \right\|_{\nu}, \tag{31}$$

where $\Pi_{\nu,\Phi_{\mathrm{rf}}}$ is defined as in (28). By definition $\Pi_{\nu,\Phi_{\mathrm{rf}}}$ is contractive under $\| \cdot \|_{\nu}$. Note that

$$Q^{\overline{\pi},\underline{\pi}}(s,a,b) = r(s,a,b) + \gamma \mathbb{E}_{\omega \sim \mathcal{N}(0,\sigma^{-2}I_d)} \left[ \phi_{\omega}(s,a,b)^{\top} \int_{\mathcal{S}} \mu_{\omega}(s') V^{\overline{\pi},\underline{\pi}}(s') ds' \right].$$

Using Hölder's inequality, along with the fact that $\|V^{\pi}\|_{\infty} = O((1-\gamma)^{-1})$, and denoting $\beta_{\omega}^{\overline{\pi},\underline{\pi}} := \int_{\mathcal{S}} \mu_{\omega}(s') V^{\overline{\pi},\underline{\pi}}(s') ds'$, we have for every $\omega$ that

$$\left\| \beta_{\omega}^{\overline{\pi},\underline{\pi}} \right\|_{\infty} = O((1-\gamma)^{-1}).$$

Additionally, recalling that $\phi_{\omega}(s,a,b) = [\psi_{\omega}(s,a,b), r(s,a,b)]$, we have

$$\sup_{((s,a,b),\omega)} |\phi_{\omega}(s,a,b)^{\top} \beta_{\omega}^{\overline{\pi},\underline{\pi}}| \leq \sup_{((s,a,b),\omega)} |\psi_{\omega}(s,a,b)|_1 \left\| \beta_{\omega}^{\overline{\pi},\underline{\pi}} \right\|_{\infty}$$

since the coordinate in $\beta_{\omega}^{\overline{\pi},\underline{\pi}}$ corresponding to the reward part of $\phi_{\omega}(s,a,b)$ is 0. Noting

$$\sup_{((s,a,b),\omega)} |\psi_{\omega}(s,a,b)|_1 \leq 2 \sup_{(s,a,b)} \frac{g_{\alpha}(f(s,a,b))}{\alpha^d},$$

we conclude that

$$\sup_{((s,a,b),\omega)} |\phi_{\omega}(s,a,b)^{\top} \beta_{\omega}^{\overline{\pi},\underline{\pi}}| \leq \sup_{((s,a,b),\omega)} |\psi_{\omega}(s,a,b)|_1 \left\| \beta_{\omega}^{\overline{\pi},\underline{\pi}} \right\|_{\infty} = O\left( \frac{\sup_{(s,a,b)} g_{\alpha}(f(s,a,b))}{\alpha^d(1-\gamma)} \right)$$

Applying Lemma D.4, we obtain the following bound:

$$\left\| Q^{\overline{\pi},\underline{\pi}} - \Pi_{\nu,\Phi_{\mathrm{rf}}} Q^{\overline{\pi},\underline{\pi}} \right\|_{\nu} = \tilde{O}\left( \frac{\gamma \sup_{s,a,b} g_{\alpha}(f(s,a,b))}{\alpha^d(1-\gamma)\sqrt{m}} \right). \tag{32}$$

Substituting (32) into (31) completes the proof. $\square$

As shown above, the approximation error for random features decreases at a rate of $O\left(m^{-1/2}\right)$ with high probability as the number of random features $m$ increases.

Next, we turn to the approximation error for Nyström features. To establish this result, we first present the following key finding.

**Lemma D.5** (cf. Lemma 8 from Ren et al. [2023]). *Consider the following Mercer decomposition (on $\mathcal{S}$) of $k_{\alpha}(\cdot,\cdot)$:*

$$k_{\alpha}(x,x') = \sum_{i=1}^{\infty} \sigma_i e_i(x) e_i(x'), \tag{33}$$

*where $\{e_i\}_{i=1}^{\infty}$ forms a countable orthonormal basis for $L_2(\mu_{\mathrm{nys}})$ with corresponding values $\{\sigma_i\}_{i=1}^{\infty}$. Let $X^{n_{\mathrm{nys}}} = \{x_i\}_{i=1}^{n_{\mathrm{nys}}}$ be an i.i.d $n_{\mathrm{nys}}$-point sample from $\mu_{\mathrm{nys}}$. In addition, let $\lambda_1 \geq \lambda_2 \geq \cdots \geq \lambda_{n_{\mathrm{nys}}}$ denote the eigenvalues of the (unnormalized) Gram matrix $K^{(n_{\mathrm{nys}})}$ in its eigende-composition $K^{(n_{\mathrm{nys}})}U = \Lambda U$ where $U^{\top}U = U^{\top} = I$. Suppose that $\sigma_j, \lambda_j/n_{\mathrm{nys}} \lesssim \exp(-\beta j^{1/h})$ for some $\beta > 0$ and $h > 0$. Suppose that $n_{\mathrm{nys}} \geq 3$ and that $\lfloor (2\log n_{\mathrm{nys}})^h/\beta^h \rfloor \leq m \leq n_{\mathrm{nys}}$.*

*Consider the rank-$m$ kernel approximation $\hat{k}_{\alpha,m}^{n_{\mathrm{nys}}}$ constructed using Nyström features, defined as follows:*

$$\hat{k}_{\alpha,m}^{n_{\mathrm{nys}}}(s,t) = \varphi_{\mathrm{nys}}(s)^{\top} \varphi_{\mathrm{nys}}(t), \tag{34}$$

*where $\varphi_{\mathrm{nys}}(\cdot) \in \mathbb{R}^m$ and is defined as*

$$(\varphi_{\mathrm{nys}})_i(\cdot) := \frac{1}{\sqrt{\lambda_i}} u_i^{\top} k_{\alpha}(X^{n_{\mathrm{nys}}},\cdot), \quad \forall i \in [m], \tag{35}$$

*where $u_i$ denotes the $i$-th column of $U$, and $k_\alpha(X^{n_{\text{nys}}}, \cdot)$ denotes a $n_{\text{nys}}$-dimensional vector where $(k_\alpha(X^{n_{\text{nys}}}, \cdot))_\ell = k_\alpha(x_\ell, \cdot)$; for details on how $\varphi_{\text{nys}}(\cdot)$ is derived, see Appendix B. Then, for any $\delta > 0$, with probability at least $1 - \delta$,*

$$\int_{\mathcal{S}} \sqrt{\left(k_\alpha - \hat{k}_{\alpha,m}^{n_{\text{nys}}}\right)(x,x)} d\mu_{\text{nys}}(x) = \tilde{O}\left(\sqrt{\sum_{i=m+1}^{n_{\text{nys}}} \frac{1}{n_{\text{nys}}} \lambda_i} + \frac{1}{n_{\text{nys}}}\right) = \tilde{O}\left(\frac{1}{n_{\text{nys}}}\right). \tag{36}$$

Leveraging Lemma D.5, we can now derive the following result for the approximation error associated with Nyström features.

**Proposition D.2** (*Q*-Approximation error with Nyström features). *Suppose all the assumptions in Lemma D.5 hold. Suppose also that we pick the sampling distribution $\mu_{\text{nys}}$ such that*

$$\mu_{\text{nys}}(x) = p_\alpha(x). \tag{37}$$

*We define the feature map $\Phi_{\text{nys}}$ for the Nyström features as follows:*

$$\Phi_{\text{nys}} = \{[\psi_{\text{nys}}(s,a,b), r(s,a,b)]\}_{(s,a,b) \in \mathcal{S} \times \mathcal{A} \times \mathcal{A}}, \tag{38}$$

*where $\psi_{\text{nys}}(s,a,b) \in \mathbb{R}^m$ is defined in (38) in Algorithm 3. Then, for any $\delta > 0$, with probability at least $1 - \delta$,*

$$\left\|Q^{\overline{\pi},\underline{\pi}} - \tilde{Q}_{\text{nys}}^{\overline{\pi},\underline{\pi}}\right\|_\nu \leq \tilde{O}\left(\frac{\gamma \tilde{g}_\alpha}{(1-\gamma)^2 n_{\text{nys}}}\right) \leq \tilde{O}\left(\frac{\gamma \tilde{g}_\alpha}{(1-\gamma)^2 m}\right),$$

*where $\|\cdot\|_\nu$ is the $L_2$ norm defined as $\|f\|_\nu = \int f^2 d\nu$, $\tilde{Q}_{\Phi_{\text{nys}}}^{\overline{\pi},\underline{\pi}}$ is defined in (26), and $\tilde{g}_\alpha := \sup_{s,a,b} \frac{g_\alpha(f(s,a,b))}{\alpha^d}$.*

*Proof.* Following the analysis at the beginning of the proof of Proposition D.1, and using the contraction property along with results from Yu and Bertsekas [2008], we have the following bound:

$$\left\|Q^{\overline{\pi},\underline{\pi}} - \tilde{Q}_{\Phi_{\text{nys}}}^{\overline{\pi},\underline{\pi}}\right\|_\nu \leq \frac{1}{1-\gamma}\left\|Q^{\overline{\pi},\underline{\pi}} - \Pi_{\nu,\Phi_{\text{nys}}} Q^{\overline{\pi},\underline{\pi}}\right\|_\nu, \tag{39}$$

where $\Pi_{\nu,\Phi_{\text{nys}}}$ is defined as in (28).

Next, we need to bound the term $\left\|Q^{\overline{\pi},\underline{\pi}} - \Pi_{\nu,\Phi_{\text{nys}}} Q^{\overline{\pi},\underline{\pi}}\right\|_\nu$. Starting from (34), and recalling the definition of $\varphi_{\text{nys}}(\cdot)$ from (35), we express our Nyström approximation of the kernel $k_\alpha\left(s', \frac{f(s,a,b)}{1-\alpha^2}\right)$ as

$$\hat{k}_m^{n_{\text{nys}}}\left(s', \frac{f(s,a,b)}{1-\alpha^2}\right) = \varphi_{\text{nys}}(s')^\top \varphi_{\text{nys}}\left(\frac{f(s,a,b)}{(1-\alpha^2)}\right). \tag{40}$$

Since we have $\psi_{\text{nys}}(s,a,b) = g_\alpha(f(s,a,b))\varphi_{\text{nys}}\left(\frac{f(s,a,b)}{1-\alpha^2}\right)$, where $\psi_{\text{nys}}(\cdot)$ is defined as in (8), this suggests the following Nyström-based approximation for the $Q$-function:

$$\hat{Q}_{\text{nys}}^{\overline{\pi},\underline{\pi}}(s,a,b) := r(s,a,b) + \gamma \psi_{\text{nys}}(s,a,b)^\top \left(\frac{\int_{\mathcal{S}} \varphi_{\text{nys}}(s') V^{\overline{\pi},\underline{\pi}}(s') p_\alpha(s') ds'}{\alpha^d}\right).$$

It is important to note that this approximation is a valid solution to the objective:

$$\underset{f \in \text{span}(\Phi_{\text{nys}})}{\arg\min} \mathbb{E}_\nu \left(Q^{\overline{\pi},\underline{\pi}}(s,a,b) - f(s,a,b)\right)^2.$$

Thus, we have

$$\left\|\Pi_{\nu,\Phi_{\mathrm{nys}}}(Q^{\overline{\pi},\pi}) - Q^{\overline{\pi},\pi}\right\|_{\nu} \le \left\|\hat{Q}^{\overline{\pi},\pi}_{\mathrm{nys}} - Q^{\overline{\pi},\pi}\right\|_{\nu}$$

$$= \gamma\left\|\psi_{\mathrm{nys}}(s,a,b)^{\top}\left(\frac{\int_{\mathcal{S}}\varphi_{\mathrm{nys}}(s')V^{\overline{\pi},\pi}(s')p_{\alpha}(s')ds'}{\alpha^d}\right)\right.$$

$$\left. - \int_{\mathcal{S}} P(s'\mid s,a,b)V^{\overline{\pi},\pi}(s')ds'\right\|_{\nu}$$

$$= \gamma\left\|\left(\frac{g_{\alpha}(f(s,a,b))}{\alpha^d}\right)\left(\int_{\mathcal{S}}\varphi_{\mathrm{nys}}\left(\frac{f(s,a,b)}{1-\alpha^2}\right)^{\top}\varphi_{\mathrm{nys}}(s')V^{\overline{\pi},\pi}(s')dp_{\alpha}(s')\right.\right.$$

$$\left.\left. - \int_{\mathcal{S}} k_{\alpha}(s',\frac{f(s,a,b)}{1-\alpha^2})V^{\overline{\pi},\pi}(s')dp_{\alpha}(s')\right)\right\|_{\nu}$$

$$\le \frac{\gamma\tilde{g}_{\alpha}}{1-\gamma}\times\underbrace{\left\|\int_{\mathcal{S}}\left|\hat{k}^{n_{\mathrm{nys}}}_m\left(s',\frac{f(s,a,b)}{1-\alpha^2}\right) - k_{\alpha}\left(s',\frac{f(s,a,b)}{1-\alpha^2}\right)\right|p_{\alpha}(s')ds'\right\|_{\nu}}_{T_1}. \qquad (41)$$

Using the one-to-one correspondence between Reproducing Kernel Hilbert Spaces (RKHS) and PD kernels, we can express the kernel function $k_{\alpha}$ as an inner product in the corresponding RKHS, $\mathcal{H}_{k_{\alpha}}$. Specifically, we have

$$k_{\alpha}\left(s',\frac{f(s,a,b)}{1-\alpha^2}\right) = \langle k_{\alpha}\left(\frac{f(s,a,b)}{1-\alpha^2},\cdot\right), k_{\alpha}(s',\cdot)\rangle_{\mathcal{H}_{k_{\alpha}}}$$

$$\hat{k}^{n_{\mathrm{nys}}}_m\left(s',\frac{f(s,a,b)}{1-\alpha^2}\right) = k_{\alpha}(\frac{f(s,a,b)}{1-\alpha^2},X^n)\sum_{i=1}^{m}\frac{1}{\lambda_i}u_iu_i^{\top}k_{\alpha}(X^n,s')$$

$$= \left\langle k_{\alpha}(\frac{f(s,a,b)}{1-\alpha^2},\cdot),k_{\alpha}(X^n,\cdot)\right\rangle_{\mathcal{H}_{k_{\alpha}}}\sum_{i=1}^{m}\frac{1}{\lambda_i}u_iu_i^{\top}k_{\alpha}(X^n,s')$$

$$= \langle k_{\alpha}\left(\frac{f(s,a,b)}{1-\alpha^2},\cdot\right),\hat{k}^{n_{\mathrm{nys}}}_m(s',)\rangle_{\mathcal{H}_{k_{\alpha}}}$$

By choosing $\mu_{\mathrm{nys}} = p_{\alpha}(x)$, continuing from (41), we have

$$T_1$$

$$= \left\|\int_{\mathcal{S}}\left|\hat{k}^{n_{\mathrm{nys}}}_m\left(s',\frac{f(s,a,b)}{1-\alpha^2}\right) - k_{\alpha}\left(s',\frac{f(s,a,b)}{1-\alpha^2}\right)\right|p_{\alpha}(s')ds'\right\|_{\nu}$$

$$= \left\|\int_{\mathcal{S}}\left|\langle k_{\alpha}\left(\frac{f(s,a,b)}{1-\alpha^2},\cdot\right),\hat{k}^{n_{\mathrm{nys}}}_m(s',)\rangle_{\mathcal{H}_{k_{\alpha}}}\right.\right.$$

$$\left.\left. - \langle k_{\alpha}\left(\frac{f(s,a,b)}{1-\alpha^2},\cdot\right),k_{\alpha}(s',\cdot)\rangle_{\mathcal{H}_{k_{\alpha}}}\right|p_{\alpha}(s')ds'\right\|_{\nu}$$

$$= \left\|\int_{\mathcal{S}}\left|\left\langle k_{\alpha}\left(\frac{f(s,a,b)}{1-\alpha^2},\cdot\right),\hat{k}^{n_{\mathrm{nys}}}_m(s',)-k_{\alpha}(s',\cdot)\right\rangle_{\mathcal{H}_{k_{\alpha}}}\right|dp_{\alpha}(s')\right\|_{\nu}$$

$$\le \left\|\int_{\mathcal{S}}\sqrt{k_{\alpha}\left(\frac{f(s,a,b)}{1-\alpha^2},\frac{f(s,a,b)}{1-\alpha^2}\right)}\sqrt{(k_{\alpha}-\hat{k}^{n_{\mathrm{nys}}}_m)(s',s')}dp_{\alpha}(s')\right\|_{\nu}$$

$$\le \left\|\int_{\mathcal{S}}\sqrt{(k_{\alpha}-\hat{k}^{n_{\mathrm{nys}}}_m)(s',s')}dp_{\alpha}(s')\right\|_{\nu}$$

To move from the second-to-last line to the last line, we used the fact that $k_{\alpha}(\cdot,\cdot)\le 1$. Next, applying Lemma D.5 and using the decay assumption on the eigenvalues from both the Mercer

expansion and the empirical Gram matrix, we obtain a probabilistic bound. Specifically, for any $\delta > 0$, with probability at least $1 - \delta$, we have

$$T_1 \leq \left( \int_{\mathcal{S}} \sqrt{(k_\alpha - \hat{k}_m^{n_{\mathrm{nys}}})(s', s')} \mu(s') ds' \right) = \tilde{O}\left( \frac{1}{n_{\mathrm{nys}}} \right).$$

Thus, by combining (41) with the bound on $T_1$, we conclude the desired result, demonstrating that the error introduced by the Nyström approximation decays at the rate $\tilde{O}(1/n_{nys})$. $\qquad \square$

*Remark* D.1. As demonstrated by the proposition above, a key advantage of the Nyström method is its ability to reduce the approximation error to $O((n_{\mathrm{nys}}^{-1}))$, where $n_{\mathrm{nys}}$ represents the number of samples used to construct the Gram matrix for generating Nyström features. This rate of $O((n_{\mathrm{nys}}^{-1}))$ consistently outperforms the $O(1/m)$ where $m$ is the number of features, since SDEPO is designed to select $m \leq n_{\mathrm{nys}}$. The only requirement for this improvement is that the eigenvalues of the empirical Gram matrix and Mercer expansion meet certain decay assumptions, which is a standard assumption to make in the kernel literature (cf. Theorem 2 in Belkin [2018]).

### D.2 Statistical Error

We now present the bound on the statistical error arising from using a finite number of samples, $n$. The result holds for both Nyström and random features.

**Proposition D.3.** *For each policy pair $\{\overline{\pi}, \underline{\pi}\}$ encountered in the algorithm, let $\hat{Q}_{\Phi,L}^{\overline{\pi},\underline{\pi}}$ denote the policy given by*

$$\hat{Q}_{\Phi,L}^{\overline{\pi},\underline{\pi}}(s, a, b) = \phi(s, a, b)^\top \hat{w}_L,$$

*where $\hat{w}_L$ is defined as in (11), and $\Phi$ can either be the reward concatenated with the Nyström or random features, i.e. either $\Phi_{\mathrm{nys}}$ or $\Phi_{\mathrm{rf}}$. Then, for sufficiently large $n$, there exists an universal constant $C > 0$ independent of $m$, $n$, $L$ and $(1 - \gamma)^{-1}$, such that with probability at least $1 - \delta$, we have*

$$\left\| \tilde{Q}_\Phi^{\overline{\pi},\underline{\pi}} - \hat{Q}_{\Phi,L}^{\overline{\pi},\underline{\pi}} \right\|_\nu \leq \gamma^L \left\| \tilde{Q}_\Phi^{\overline{\pi},\underline{\pi}} \right\|_\nu + \frac{C \tilde{g}_\alpha^6 m^3 \mathrm{polylog}(m, L/\delta)}{(1 - \gamma) \Upsilon_1^2 \Upsilon_2 \sqrt{n}}, \tag{42}$$

*where we recall $\tilde{Q}_\Phi^{\overline{\pi},\underline{\pi}}$ is defined as in (26).*

*Proof.* To simplify the presentation, let us define $\Phi$ as the concatenation of $\phi(s, a, b)$ across all $(s, a, b) \in \mathcal{S} \times \mathcal{A} \times \mathcal{A}$, and define the operator $P^{\overline{\pi},\underline{\pi}}$ as

$$(P^{\overline{\pi},\underline{\pi}} f)(s, a, b) = \mathbb{E}_{(s',a')\sim P(s,a,b) \times \overline{\pi} \times \underline{\pi}} f(s', a', b').$$

Additionally, define $\tilde{w}$ as the solution to the equation:

$$\tilde{w} = \left( \mathbb{E}_\nu \left[ \phi(s, a, b) \phi(s, a, b)^\top \right] \right)^{-1}$$
$$\left( \mathbb{E}_\nu \left[ \phi(s, a, b) \left( r(s, a, b) + \gamma \mathbb{E}_{(s',a',b')\sim P(s,a,b) \times \overline{\pi} \times \underline{\pi}} \left[ \phi(s', a', b')^\top \tilde{w} \right] \right) \right] \right),$$

and let $\tilde{Q}(s, a, b) = \phi(s, a, b)^\top \tilde{w}$. It is straightforward to observe that $\tilde{w}$ is the fixed point of the population (*i.e.*, $n \to \infty$) projected least square update (11). Furthermore, we can express $\tilde{w}$ as

$$\tilde{w} = \left( \mathbb{E}_\nu \left[ \phi(s, a, b) \left( \phi(s, a, b) - \gamma \mathbb{E}_{(s',a',b')\sim P(s,a,b) \times \overline{\pi} \times \underline{\pi}} \phi(s', a', b') \right)^\top \right] \right)^{-1}$$
$$\mathbb{E}_\nu \left[ \phi(s, a, b) r(s, a, b) \right].$$

For the sake of brevity, we will omit the subscript when $\nu$ refers to the Lebesgue measure. We define $\hat{\Pi}_\nu$ and $\hat{P}^{\overline{\pi},\underline{\pi}}$ as the empirical counterparts of $\Pi_\nu$ and $P^{\overline{\pi},\underline{\pi}}$, respectively. Under the update rule (11), we have the following relations:

$$\Phi \hat{w}_{t+1} = \hat{\Pi}_\nu (r + \gamma \hat{P}^{\overline{\pi},\underline{\pi}} \Phi \hat{w}_t),$$
$$\Phi \tilde{w} = \Pi_\nu (r + \gamma P^{\overline{\pi},\underline{\pi}} \Phi \tilde{w}),$$

which lead to

$$\Phi(\tilde{w} - \hat{w}_{t+1}) = (\Pi_\nu - \hat{\Pi}_\nu)r + \gamma(\Pi_\nu P^{\overline{\pi},\underline{\pi}})\Phi(\tilde{w} - \hat{w}_t) + \gamma(\Pi_\nu P^{\overline{\pi},\underline{\pi}} - \hat{\Pi}_\nu \hat{P}^{\overline{\pi},\underline{\pi}})\Phi\hat{w}_t.$$

Applying the triangle inequality, we can bound this as:

$$\left\|\Phi(\tilde{w} - \hat{w}_{t+1})\right\|_\nu \leq \gamma\left\|\Phi(\tilde{w} - \hat{w}_t)\right\|_\nu + \left\|\left(\Pi_\nu - \hat{\Pi}_\nu\right)r\right\|_\nu + \gamma\left\|\left(\Pi_\nu P^{\overline{\pi},\underline{\pi}} - \hat{\Pi}_\nu \hat{P}^{\overline{\pi},\underline{\pi}}\right)\Phi\hat{w}_t\right\|_\nu,$$

where we use the contractivity under $\|\cdot\|_\nu$. Telescoping over $t$, we have

$$\left\|\Phi(\tilde{w} - \hat{w}_L)\right\|_\nu \leq \gamma^T\left\|\Phi(\tilde{w} - \hat{w}_0)\right\|_\nu + \frac{1}{1-\gamma}\left\|\left(\Pi_\nu - \hat{\Pi}_\nu\right)r\right\|_\nu$$
$$+ \frac{\gamma}{1-\gamma}\max_{l\in[L]}\left\|\left(\Pi_\nu P^{\overline{\pi},\underline{\pi}} - \hat{\Pi}_\nu \hat{P}^{\overline{\pi},\underline{\pi}}\right)\Phi\hat{w}_t\right\|_\nu.$$

The following proof follows the same reasoning as Appendix E in Ren et al. [2023]. □

Proposition D.3 shows that the statistical error of the linear components can be broken down into two parts. The first part, $\gamma^L\left\|\tilde{Q}^{\overline{\pi},\underline{\pi}}\right\|_\nu$, arises from initializing with $\hat{w}_0 = 0$ and decreases as the number of least-squares policy evaluation iterations $L$ increases. The second part, $\frac{C\tilde{g}_\alpha^6 m^3 \mathrm{polylog}(m,L/\delta)}{(1-\gamma)\Upsilon_1^2\Upsilon_2\sqrt{n}}$, represents the statistical error from using a finite sample size and diminishes as $n$ increases. By choosing $L = \Theta(\log n)$, we can balance both parts, resulting in an estimation error that decreases at a rate of $O(n^{-1/2})$ with high probability.

### D.3 Total Error for Policy Evaluation

Combine the approximation error in Proposition D.1 and Proposition D.2 with the statistical error from Proposition D.3, we obtain the following bound on the total error for least-squares policy evaluation:

**Theorem D.1** (i.e. Theorem D.1). *Let $L = \Theta(\log n)$. With probability at least $1 - \delta$, we have that for the random features $\Phi_{\mathrm{rf}}$,*

$$\left\|Q^{\overline{\pi},\underline{\pi}} - \hat{Q}_{\Phi_{\mathrm{rf}},L}^{\overline{\pi},\underline{\pi}}\right\|_\nu = \tilde{O}\left(\underbrace{\frac{\tilde{g}_\alpha}{(1-\gamma)^2\sqrt{m}}}_{\text{approx. error}} + \underbrace{\frac{\tilde{g}_\alpha^6 m^3}{(1-\gamma)\Upsilon_1^2\Upsilon_2\sqrt{n}}}_{\text{stat. error}}\right), \tag{43}$$

*and for the Nyström features*

$$\left\|Q^{\overline{\pi},\underline{\pi}} - \hat{Q}_{\Phi_{\mathrm{nys}},L}^{\overline{\pi},\underline{\pi}}\right\|_\nu = \tilde{O}\left(\underbrace{\frac{\tilde{g}_\alpha}{(1-\gamma)^2 n_{\mathrm{nys}}}}_{\text{approx. error}} + \underbrace{\frac{\tilde{g}_\alpha^6 m^3}{(1-\gamma)\Upsilon_1^2\Upsilon_2\sqrt{n}}}_{\text{stat. error}}\right).$$

*Proof.* This result follows directly from the triangle inequality, *i.e.*,

$$\left\|Q^{\overline{\pi},\underline{\pi}} - \hat{Q}_{\Phi,L}^{\overline{\pi},\underline{\pi}}\right\|_\nu \leq \left\|Q^{\overline{\pi},\underline{\pi}} - \tilde{Q}_\Phi^{\overline{\pi},\underline{\pi}}\right\|_\nu + \left\|\tilde{Q}_\Phi^{\overline{\pi},\underline{\pi}} - \hat{Q}_{\Phi,L}^{\overline{\pi},\underline{\pi}}\right\|_\nu.$$

The first term, representing the approximation error, is bounded by Proposition D.1 and Proposition D.2 for random and Nyström features, respectively. The second term, capturing the statistical error, is bounded by Proposition D.3. □

Theorem D.1 provides the estimation error bound for the $Q$-function using least-squares policy evaluation, applicable to both random and Nyström features. The result reveals a key tradeoff between the approximation error and statistical error. For random features, increasing the number of features $m$ enhances the ability to approximate the original infinite-dimensional function space, as reflected by the $\tilde{O}\left(\frac{1}{\sqrt{m}}\right)$ approximation error term on the RHS of (43). However, this improvement comes at

the cost of needing more learning samples $n$ during policy evaluation to effectively train the weights, as indicated by the $\tilde{O}\left(\frac{m^3}{\sqrt{n}}\right)$ statistical error term in (43). For Nyström features, this tradeoff may be better especially since the approximation term scales as $\tilde{O}(1/n_{\text{nys}}) \leq \tilde{O}(1/m)$ while the statistical error term remains the same as with random features. As a result, the Nyström method offers a better approximation error rate with additional mild assumption.

## E  Convergence analysis of `SDEPO`

In this section, we denote $Q_{k-1} = Q^{\overline{\pi}_{k-1}, \underline{\pi}_{k-1}}$, and use the notation $\overline{\pi}^s Q^s \underline{\pi}^s = \mathbb{E}_{a \sim \overline{\pi}(\cdot|s), b \sim \underline{\pi}(\cdot|s)}[Q(s,a,b)]$.

To establish the convergence of the policy optimization procedure, we follow the error propagation framework from Perolat et al. [2015], which consists of two key stages:

Stage 1: Identify an approximate solution $\underline{\pi}_k$ such that

$$\mathbb{E} \max_{\overline{\pi}} \overline{\pi}^s Q_{k-1}^s \underline{\pi}_k^s - \min_{\underline{\pi}} \max_{\overline{\pi}} \overline{\pi}^s Q_{k-1}^s \underline{\pi}^s = \epsilon_1^k(s),$$

where the expectation is taken over the randomness of the algorithm used to generate $\underline{\pi}_k$.

In our analysis, we bound the stronger quantity, which is called duality gap

$$\mathbb{E}\left[\max_{\overline{\pi}} \overline{\pi}^s Q_{k-1}^s \underline{\pi}_k^s - \min_{\underline{\pi}} \overline{\pi}_k^s Q_{k-1}^s \underline{\pi}^s\right] \geq \epsilon_1^k(s),$$

where the bound follows from the definition of Nash equilibrium, as $\min_{\underline{\pi}} \overline{\pi}_k Q_{k-1} \underline{\pi} \leq \min_{\underline{\pi}} \max_{\overline{\pi}} \overline{\pi} Q_{k-1} \underline{\pi}$.

Stage 2: Identify an approximate solution $\overline{\pi}_k$ such that

$$V^{\overline{\pi}_k, \underline{\pi}_k^*}(s) - \mathbb{E} V^{\overline{\pi}_k, \underline{\pi}_k}(s) = \epsilon_2^k(s),$$

where the expectation is over the randomness of the algorithm used to generate $\overline{\pi}_k$.

Following the analysis in Perolat et al. [2015], we conclude that there exists a constant $C$, such that

$$\mathbb{E}_{s \sim \mu_0}\left[\max_{\overline{\pi}} V^{\overline{\pi}, \underline{\pi}_k}(s) - V^*(s)\right] \leq C\left(\frac{k}{1-\gamma} \sup_{j \in 1, \cdots, k-1} \mathbb{E}_s[\epsilon_2^j(s) + \epsilon_2^j(s)] + \frac{\gamma^k}{1-\gamma}\right). \quad (44)$$

First, we analyze the error in the Stage 1 of Algorithm 3.

**Lemma E.6** (i.e. Lemma 3). *Let Assumption 5 hold. In Stage 1 of Algorithm 3,*

$$\mathbb{E}\left[\max_{\overline{\pi}} \overline{\pi} Q_{k-1} \underline{\pi}_k - \min_{\underline{\pi}} \overline{\pi}_k Q_{k-1} \underline{\pi}\right]$$

$$\leq \frac{1}{\eta T} \log \frac{1}{\underline{c}} + 2\|\hat{Q}^{\overline{\pi}_{k-1}, \underline{\pi}_{k-1}} - Q^{\overline{\pi}_{k-1}, \underline{\pi}_{k-1}}\|_{\nu_{\overline{\pi}_{k-1}, \underline{\pi}_{k-1}}} + \frac{\eta}{2(1-\gamma)^2}.$$

*Proof.* Denote $Q = Q_{k-1}$. Recall the notation $\overline{\pi}^s Q^s \underline{\pi}^s = \mathbb{E}_{a \sim \overline{\pi}(\cdot|s), b \sim \underline{\pi}(\cdot|s)}[Q(s,a,b)]$. First by definition of $\underline{\pi}_k$ and the standard formulation for the duality gap, we have for all $s$

$$\overline{\pi}^s Q^s \underline{\pi}_k - \overline{\pi}_k^s Q^s \underline{\pi}^s = \frac{1}{T} \sum_{t=1}^{T} \langle \mathbb{E}_{b \sim \underline{\pi}_{k,t}(\cdot|s)} Q(s,a,\cdot), \overline{\pi}(\cdot|s) \rangle - \frac{1}{T} \sum_{t=1}^{T} \langle \mathbb{E}_{a \sim \overline{\pi}_{k,t}(\cdot|s)} Q(s,\cdot,b), \underline{\pi}(\cdot|s) \rangle$$

$$= \frac{1}{T} \sum_{t=1}^{T} \langle \mathbb{E}_{b \sim \underline{\pi}_{k,t}(\cdot|s)} Q(s,a,\cdot), \overline{\pi}(\cdot|s) - \overline{\pi}_{k,t}(\cdot|s) \rangle$$

$$- \langle \mathbb{E}_{a \sim \overline{\pi}_{k,t}(\cdot|s)} Q(s,\cdot,b), \underline{\pi}(\cdot|s) - \underline{\pi}_{k,t}(\cdot|s) \rangle.$$

From the update rule of $\underline{\pi}_{k,t+1}$, it holds for all $k, s, \underline{\pi}$ that

$$\langle \nabla KL(\underline{\pi}_{k,t+1}, \underline{\pi}_{k,t}) + \eta \mathbb{E}_{a \sim \overline{\pi}_{k,t}}[\hat{Q}^{\overline{\pi}_{k-1}, \underline{\pi}_{k-1}}(s,a,\cdot)], \underline{\pi}(\cdot|s) - \underline{\pi}_{k,t+1}(\cdot|s) \rangle \geq 0.$$

Applying the three-point identity yields:

$$KL(\underline{\pi}(\cdot|s), \underline{\pi}_{k,t+1}(\cdot|s)) \leq KL(\underline{\pi}(\cdot|s), \underline{\pi}_{k,t}(\cdot|s)) - KL(\underline{\pi}_{k,t+1}(\cdot|s), \underline{\pi}_{k,t}(\cdot|s))$$

$$+\eta\langle\mathbb{E}_{a\sim\overline{\pi}_{k-1}}[\hat{Q}^{\overline{\pi}_{k-1},\underline{\pi}_{k-1}}(s,a,\cdot) - Q^{\overline{\pi}_{k-1},\underline{\pi}_{k-1}}(s,a,\cdot)], \underline{\pi}(\cdot|s) - \underline{\pi}_{k,t+1}(\cdot|s)\rangle$$

$$+\eta\langle\mathbb{E}_{a\sim\overline{\pi}_{k-1}}[Q_{k-1}(s,a,\cdot)], \underline{\pi}(\cdot|s) - \underline{\pi}_{k,t+1}(\cdot|s)\rangle.$$

We can bound the inner products using Cauchy-Schwarz, Young's and Pinsker's inequalities,

$$\eta\langle\mathbb{E}_{a\sim\overline{\pi}_{k-1}}[\hat{Q}^{\overline{\pi}_{k-1},\underline{\pi}_{k-1}}(s,a,\cdot) - Q^{\overline{\pi}_{k-1},\underline{\pi}_{k-1}}(s,a,\cdot)], \underline{\pi}(\cdot|s) - \underline{\pi}_{k,t+1}(\cdot|s)\rangle$$

$$\leq 2\eta\|\hat{Q}^{\overline{\pi}_{k-1},\underline{\pi}_{k-1}} - Q^{\overline{\pi}_{k-1},\underline{\pi}_{k-1}}\|_{\nu_{\overline{\pi}_{k-1},\underline{\pi}_{k-1}}}$$

$$\eta\langle\mathbb{E}_{a\sim\overline{\pi}_{k-1}}[Q^{\overline{\pi}_{k-1},\underline{\pi}_{k-1}}(s,a,\cdot)], \underline{\pi}(\cdot|s) - \underline{\pi}_{k,t+1}(\cdot|s)\rangle$$

$$=\eta\langle\mathbb{E}_{a\sim\overline{\pi}_{k-1}}[Q^{\overline{\pi}_{k-1},\underline{\pi}_{k-1}}(s,a,\cdot)], \underline{\pi}(\cdot|s) - \underline{\pi}_{k,t}(\cdot|s)\rangle$$

$$+\eta\langle\mathbb{E}_{a\sim\overline{\pi}_{k-1}}[Q^{\overline{\pi}_{k-1},\underline{\pi}_{k-1}}(s,a,\cdot)], \underline{\pi}_{k,t}(\cdot|s) - \underline{\pi}_{k,t+1}(\cdot|s)\rangle$$

$$\leq\eta\langle\mathbb{E}_{a\sim\overline{\pi}_{k-1}}[Q^{\overline{\pi}_{k-1},\underline{\pi}_{k-1}}(s,a,\cdot)], \underline{\pi}(\cdot|s) - \underline{\pi}_{k,t}(\cdot|s)\rangle + \frac{\eta^2}{2(1-\gamma)^2} + KL(\underline{\pi}_{k,t+1}(\cdot|s), \underline{\pi}_{k,t}(\cdot|s))$$

Combining these estimates gives us:

$$\langle\mathbb{E}_{a\sim\overline{\pi}_{k-1}}[Q^{\overline{\pi}_{k-1},\underline{\pi}_{k-1}}(s,a,\cdot)], \underline{\pi}_{k,t}(\cdot|s) - \underline{\pi}(\cdot|s)\rangle + \frac{1}{\eta}KL(\underline{\pi}(\cdot|s), \underline{\pi}_{k,t+1}(\cdot|s))$$

$$\leq\frac{1}{\eta}KL(\underline{\pi}(\cdot|s), \underline{\pi}_{k,t}(\cdot|s)) + 2\|\hat{Q}^{\overline{\pi}_{k-1},\underline{\pi}_{k-1}} - Q^{\overline{\pi}_{k-1},\underline{\pi}_{k-1}}\|_{\nu_{\overline{\pi}_{k-1},\underline{\pi}_{k-1}}} + \frac{\eta^2}{2(1-\gamma)^2}$$

Summing these inequalities results in:

$$\frac{1}{T}\sum_{t=1}^{T}\langle\mathbb{E}_{a\sim\overline{\pi}_{k,t}}[Q^{\overline{\pi}_{k,t},\underline{\pi}_{k,t}}(s,a,\cdot)], \underline{\pi}_{k,t}(\cdot|s) - \underline{\pi}(\cdot|s)\rangle$$

$$\leq\frac{1}{\eta T}KL(\underline{\pi}, \underline{\pi}_{k,0}) + 2\|\hat{Q}^{\overline{\pi}_{k-1},\underline{\pi}_{k-1}} - Q^{\overline{\pi}_{k-1},\underline{\pi}_{k-1}}\|_{\nu_{\overline{\pi}_{k-1},\underline{\pi}_{k-1}}} + \frac{\eta^2}{2(1-\gamma)^2}$$

$$\leq\frac{1}{\eta T}\log\frac{1}{\underline{c}} + 2\|\hat{Q}^{\overline{\pi}_{k-1},\underline{\pi}_{k-1}} - Q^{\overline{\pi}_{k-1},\underline{\pi}_{k-1}}\|_{\nu_{\overline{\pi}_{k-1},\underline{\pi}_{k-1}}} + \frac{\eta}{2(1-\gamma)^2}$$

$$\square$$

This result shows how the error in the approximation of the $Q$-function and the learning rate $\eta$ impact the optimization performance in Stage 1. Next, we turn to Stage 2 and provide an upper bound on the one-sided error.

**Lemma E.7** (i.e. Lemma 4). *Let Assumption 5 hold and $\mu_0$ be a state distribution. In Stage 2 of Algorithm 3,*

$$\frac{1}{T}\sum_{t=1}^{T}\mathbb{E}_{s\sim\mu_0}\left[V^{\overline{\pi}_k^*,\underline{\pi}_k}(s) - V^{\overline{\pi}'_{k,t},\underline{\pi}_k}(s)\right]$$

$$\leq\frac{1}{1-\gamma}\left(\frac{\log\frac{1}{\underline{c}}}{\eta T} + \frac{\eta}{(1-\gamma)^2} + \eta\|\hat{Q}^{\overline{\pi}'_{k,t},\underline{\pi}_k} - Q^{\overline{\pi}'_{k,t},\underline{\pi}_k}\|_{\nu_{\overline{\pi}'_{k,t},\underline{\pi}_k}}\right).$$

*Proof.* By the update rule for $\overline{\pi}'_{k,t+1}$, for all $s$ and $\overline{\pi}$, we have that

$$KL(\overline{\pi}(\cdot|s), \overline{\pi}'_{k,t+1}(\cdot|s)) \leq KL(\overline{\pi}(\cdot|s), \overline{\pi}'_{k,t}(\cdot|s))$$

$$- \eta\langle\mathbb{E}_{b\sim\underline{\pi}_k}[\hat{Q}^{\overline{\pi}'_{k,t},\underline{\pi}_k}(s,\cdot,b)], \overline{\pi}(\cdot|s) - \overline{\pi}'_{k,t+1}(\cdot|s)\rangle - KL(\overline{\pi}'_{k,t+1}(\cdot|s), \overline{\pi}'_{k,t}(\cdot|s))$$

$$= KL(\overline{\pi}(\cdot|s), \overline{\pi}'_{k,t}(\cdot|s)) - \eta\langle\mathbb{E}_{b\sim\underline{\pi}_k}[\hat{Q}^{\overline{\pi}'_{k,t},\underline{\pi}_k}(s,\cdot,b) - Q^{\overline{\pi}'_{k,t},\underline{\pi}_k}(s,\cdot,b)], \overline{\pi}(\cdot|s) - \overline{\pi}'_{k,t+1}(\cdot|s)\rangle$$

$$- \eta\langle\mathbb{E}_{b\sim\underline{\pi}_k}[Q^{\overline{\pi}'_{k,t},\underline{\pi}_k}(s,\cdot,b)], \overline{\pi}(\cdot|s) - \overline{\pi}'_{k,t+1}(\cdot|s)\rangle - KL(\overline{\pi}'_{k,t+1}(\cdot|s), \overline{\pi}'_{k,t}(\cdot|s))$$

We can bound the inner products using Cauchy-Schwarz, Young's and Pinsker's inequalities

$$\eta\langle\mathbb{E}_{b\sim\underline{\pi}_k}[\hat{Q}^{\overline{\pi}'_{k,t},\underline{\pi}_k}(s,\cdot,b)-Q^{\overline{\pi}'_{k,t},\underline{\pi}_k}(s,\cdot,b)],\overline{\pi}(\cdot|s)-\overline{\pi}'_{k,t+1}(\cdot|s)\rangle$$
$$\leq O(\eta\|\hat{Q}^{\overline{\pi}'_{k,t},\underline{\pi}_k}-Q^{\overline{\pi}'_{k,t},\underline{\pi}_k}\|_{\nu_{\overline{\pi}'_{k,t},\underline{\pi}_k}})$$

and

$$\eta\langle\mathbb{E}_{b\sim\underline{\pi}_k}[Q^{\overline{\pi}'_{k,t},\underline{\pi}_k}(s,\cdot,b)],\overline{\pi}(\cdot|s)-\overline{\pi}'_{k,t+1}(\cdot|s)\rangle$$
$$=-\eta\langle\mathbb{E}_{b\sim\underline{\pi}_k}[Q^{\overline{\pi}'_{k,t},\underline{\pi}_k}(s,\cdot,b)],\overline{\pi}(\cdot|s)-\overline{\pi}'_{k,t}(\cdot|s)\rangle$$
$$-\eta\langle\mathbb{E}_{b\sim\underline{\pi}_k}[Q^{\overline{\pi}'_{k,t},\underline{\pi}_k}(s,\cdot,b)],\overline{\pi}'_{k,t}(\cdot|s)-\overline{\pi}'_{k,t+1}(\cdot|s)\rangle$$
$$\leq-\eta\langle\mathbb{E}_{b\sim\underline{\pi}_k}[Q^{\overline{\pi}'_{k,t},\underline{\pi}_k}(s,\cdot,b)],\overline{\pi}(\cdot|s)-\overline{\pi}'_{k,t}(\cdot|s)\rangle$$
$$+\frac{\eta^2\|Q^{\overline{\pi}'_{k,t},\underline{\pi}_k}(s,a,b)\|_\infty^2}{2}+KL(\overline{\pi}'_{k,t+1}(\cdot|s),\overline{\pi}'_{k,t}(\cdot|s))$$

Consequently, we have that

$$\langle\mathbb{E}_{b\sim\underline{\pi}_k}[Q^{\overline{\pi}'_{k,t},\underline{\pi}_k}(s,\cdot,b)],\overline{\pi}(\cdot|s)-\overline{\pi}'_{k,t}(\cdot|s)\rangle+\frac{1}{\eta}KL(\overline{\pi}(\cdot|s),\overline{\pi}'_{k,t+1}(\cdot|s))|$$

$$\leq\frac{1}{\eta}KL(\overline{\pi},\overline{\pi}'_{k,t})+O(\eta(\|Q^{\overline{\pi}'_{k,t},\underline{\pi}_k}(s,a,b)\|_\infty^2+\|\hat{Q}^{\overline{\pi}'_{k,t},\underline{\pi}_k}-Q^{\overline{\pi}'_{k,t},\underline{\pi}_k}\|_{\nu_{\overline{\pi}'_{k,t},\underline{\pi}_k}}))$$

Summing the inequality, we get

$$\frac{1}{T}\sum_{t=0}^{T}\langle\mathbb{E}_{b\sim\underline{\pi}_k}[Q^{\overline{\pi}'_{k,t},\underline{\pi}_k}(s,\cdot,b)],\overline{\pi}(\cdot|s)-\overline{\pi}'_{k,t}(\cdot|s)\rangle \tag{45}$$
$$\leq\frac{1}{\eta T}KL(\overline{\pi},\overline{\pi}'_{k,t})+\frac{\eta}{(1-\gamma)^2}+\eta\|\hat{Q}^{\overline{\pi}'_{k,t},\underline{\pi}_k}-Q^{\overline{\pi}'_{k,t},\underline{\pi}_k}\|_{\nu_{\overline{\pi}'_{k,t},\underline{\pi}_k}} \tag{46}$$

By the performance difference lemma, we obtain

$$V^{\overline{\pi}_k^*,\underline{\pi}_k}(s_0)-V^{\overline{\pi}'_{k,t},\underline{\pi}_k}(s_0)=\frac{1}{1-\gamma}\mathbb{E}_{s\sim d_{s_0}^{\overline{\pi}_k^*,\underline{\pi}_k}}\langle\mathbb{E}_{b\sim\underline{\pi}_k}(\cdot|s)Q^{\overline{\pi}'_{k,t},\underline{\pi}_k}(s,\cdot,b),\overline{\pi}_k^*(\cdot|s)-\overline{\pi}'_{k,t}(\cdot|s)\rangle.$$

$\square$

**Proposition E.4.** *Let Assumption 5 hold and $\mu_0$ be a state distribution,*

$$E_{s\sim\mu_0}\left[\max_{\overline{\pi}}V^{\overline{\pi},\underline{\pi}_K}(s)-V^*(s)\right]$$

$$\leq O\left(\frac{CK}{(1-\gamma)^2}(\frac{\log\frac{1}{\underline{c}}}{\eta T}+\frac{\eta}{(1-\gamma)^2}+\eta\|\hat{Q}^{\overline{\pi}_{k-1},\underline{\pi}_{k-1}}-Q^{\overline{\pi}_{k-1},\underline{\pi}_{k-1}}\|_{\nu_{\overline{\pi}_{k-1},\underline{\pi}_{k-1}}}\right.$$

$$\left.+\eta\|\hat{Q}^{\overline{\pi}'_{k,t},\underline{\pi}_k}-Q^{\overline{\pi}'_{k,t},\underline{\pi}_k}\|_{\nu_{\overline{\pi}'_{k,t},\underline{\pi}_k}})+\frac{\gamma^K}{1-\gamma}\right).$$

*where $C$ is a problem-dependent constant.*

*Proof.* Inserting the results of Lemma E.6,Lemma E.7 to (44) gives the result. $\square$

Let $L=\Theta(\log n)$ and $T=\sqrt{\frac{\log 1/\underline{c}}{\log n}(\frac{1}{(1-\gamma)^2}+\frac{\tilde{g}_\alpha}{(1-\gamma)^2\sqrt{m}}+\frac{\tilde{g}_\alpha^6 m^3}{(1-\gamma)\Upsilon_1^2\Upsilon_2\sqrt{n}})}$ for random features and $T=\sqrt{\frac{\log 1/\underline{c}}{\log n}(\frac{1}{(1-\gamma)^2}+\frac{\tilde{g}_\alpha}{(1-\gamma)^2 n_{\text{nys}}}+\frac{\tilde{g}_\alpha^6 m^3}{(1-\gamma)\Upsilon_1^2\Upsilon_2\sqrt{n}})}$ for Nyström features. Combining the error of policy evaluation in Theorem D.1, we have that the iteration complexity to get $E_{s\sim\mu_0}\left[\max_{\overline{\pi}}V^{\overline{\pi},\underline{\pi}_k}(s)-V^*(s)\right]\leq\epsilon$ is $\widetilde{O}(\frac{1}{(1-\gamma)^3\epsilon})$.

## E.1 proof with $\beta$-greedy exploration

In this section, we are going to use $\beta-$greedy policy to avoid Assumption 5. Let us define the modified policies with greedy exploration

$$\hat{\overline{\pi}} = (1-\beta)\overline{\pi} + \beta Unif(A), \quad \hat{\underline{\pi}} = (1-\beta)\underline{\pi} + \beta Unif(A)$$

Now we are going to sample with the $\hat{\overline{\pi}}, \hat{\underline{\pi}}$ and the algorithm will read as Algorithm 4.

**Lemma E.8.** *In Stage 1 of Algorithm 4,*

$$\mathbb{E}\left[\max_{\overline{\pi}} \overline{\pi} Q_{k-1}\underline{\pi}_k - \min_{\underline{\pi}} \overline{\pi}_k Q_{k-1}\underline{\pi}\right]$$

$$\leq \frac{\eta}{2(1-\gamma)^2} + \frac{32\gamma\beta}{(1-\gamma)^2} + \frac{1}{\eta T}\log\frac{1}{|\mathcal{A}|} + 2\eta\|\hat{Q}^{\hat{\overline{\pi}}_{k-1},\hat{\underline{\pi}}_{k-1}} - \hat{Q}\|_{\nu_{\hat{\overline{\pi}}_{k-1},\hat{\underline{\pi}}_{k-1}}}.$$

*Proof.* Denote $Q = Q_{k-1}, \hat{Q} = Q^{\hat{\overline{\pi}}_{k-1},\hat{\underline{\pi}}_{k-1}}$. First by definition, it holds for all $s$

$$\overline{\pi}^s Q^s \underline{\pi}_k^s - \overline{\pi}_k^s Q^s \underline{\pi}^s$$

$$=\frac{1}{T}\sum_{t=1}^{T}\left[\langle\mathbb{E}_{b\sim\underline{\pi}_{k,t}(\cdot|s)}Q(s,\cdot,b), \overline{\pi}(\cdot|s) - \overline{\pi}_{k,t}(\cdot|s)\rangle - \langle\mathbb{E}_{a\sim\overline{\pi}_{k,t}(\cdot|s)}Q(s,a,\cdot), \underline{\pi}(\cdot|s) - \underline{\pi}_{k,t}(\cdot|s)\rangle\right]$$

$$=\frac{1}{T}\sum_{t=1}^{T}\left[\langle\mathbb{E}_{b\sim\underline{\pi}_{k,t}(\cdot|s)}\hat{Q}(s,\cdot,b), \overline{\pi}(\cdot|s) - \overline{\pi}_{k,t}(\cdot|s)\rangle - \langle\mathbb{E}_{a\sim\overline{\pi}_{k,t}(\cdot|s)}\hat{Q}(s,a,\cdot), \underline{\pi}(\cdot|s) - \underline{\pi}_{k,t}(\cdot|s)\rangle\right]$$

$$+\frac{1}{T}\sum_{t=1}^{T}\left[\langle\mathbb{E}_{b\sim\underline{\pi}_{k,t}(\cdot|s)}[Q(s,\cdot,b) - \hat{Q}(s,\cdot,b)], \overline{\pi}(\cdot|s) - \overline{\pi}_{k,t}(\cdot|s)\rangle\right.$$

$$\left. - \langle\mathbb{E}_{a\sim\overline{\pi}_{k,t}(\cdot|s)}[Q(s,a,\cdot) - \hat{Q}(s,a,\cdot)], \underline{\pi}(\cdot|s) - \underline{\pi}_{k,t}(\cdot|s)\rangle\right]$$

For the error terms note

$$\langle\mathbb{E}_{b\sim\underline{\pi}_{k,t}(\cdot|s)}[Q(s,\cdot,b) - \hat{Q}(s,\cdot,b)], \overline{\pi}(\cdot|s) - \overline{\pi}_t(\cdot|s)\rangle$$

$$\leq 2\|\mathbb{E}_{b\sim\underline{\pi}_{k,t}(\cdot|s)}Q(s,\cdot,b) - \hat{Q}(s,\cdot,b)\|_\infty$$

$$\leq 2\gamma\max_{a,b}|\mathbb{E}_{s'\sim P(\cdot|s,a,b)}[V^{\overline{\pi}_{k-1},\underline{\pi}_{k-1}}(s') - V^{\hat{\overline{\pi}}_{k-1},\hat{\underline{\pi}}_{k-1}}(s')]|$$

$$\leq 2\gamma\|V^{\overline{\pi}_{k-1},\underline{\pi}_{k-1}} - V^{\hat{\overline{\pi}}_{k-1},\hat{\underline{\pi}}_{k-1}}\|_\infty$$

$$\leq \frac{16\gamma\beta}{(1-\gamma)^2},$$

where the last step is due to the Lipschitzness of the value function due to performance difference lemma and that the policies $\hat{\overline{\pi}}_{k-1}, \hat{\underline{\pi}}_{k-1}$ and $\overline{\pi}_{k-1}, \underline{\pi}_{k-1}$ differ at most by $\beta$.

By three point identity, we have

$$KL(\underline{\pi}(\cdot|s), \underline{\pi}_{k,t+1}(\cdot|s)) \leq KL(\underline{\pi}(\cdot|s), \underline{\pi}_{k,t}(\cdot|s))$$

$$+ \eta\langle\mathbb{E}_{a\sim\hat{\overline{\pi}}_{k-1}}[\hat{Q}^{\hat{\overline{\pi}}_{k-1},\hat{\underline{\pi}}_{k-1}}(s,a,\cdot) - \hat{Q}(s,a,\cdot)], \underline{\pi}(\cdot|s) - \underline{\pi}_{k,t+1}(\cdot|s)\rangle$$

$$+ \eta\langle\mathbb{E}_{a\sim\hat{\overline{\pi}}_{k,t}}[\hat{Q}(s,a,\cdot)], \underline{\pi}(\cdot|s) - \underline{\pi}_{k,t+1}(\cdot|s)\rangle$$

$$- KL(\underline{\pi}_{k,t+1}(\cdot|s), \underline{\pi}_{k,t}(\cdot|s)).$$

Again we use Cauchy-Schwarz and Youngss inequalities

$$\eta\langle\mathbb{E}_{a\sim\hat{\overline{\pi}}_{k-1}}[\hat{Q}^{\hat{\overline{\pi}}_{k-1},\hat{\underline{\pi}}_{k-1}}(s,a,\cdot) - \hat{Q}(s,a,\cdot)], \underline{\pi}(\cdot|s) - \underline{\pi}_{k,t+1}(\cdot|s)\rangle \leq 2\eta\|\hat{Q}^{\hat{\overline{\pi}}_{k-1},\hat{\underline{\pi}}_{k-1}} - \hat{Q}\|_{\nu_{\hat{\overline{\pi}}_{k-1},\hat{\underline{\pi}}_{k-1}}}$$

**Algorithm 4** Spectral Dynamic Embedding Policy Optimization with $\beta$-greedy exploration

---

**Data:** Transition Model $s' = f(s, a, b) + \varepsilon$ where $\varepsilon \sim \mathcal{N}(0, \sigma^2 I_d)$, Reward Function $r(s, a, b)$, Number of Random/Nyström Feature $m$, Number of Nyström Samples $n_{\mathrm{Nys}} \geq m$, Nyström Sampling Distribution $\mu_{\mathrm{Nys}}$, Number of Sample $n$, Factorization Scale $\alpha$, Learning Rate $\eta$

**Result:** $\pi_k$

23 Generate $\phi(s, a, b)$ using Algorithm 1 or Algorithm 2.

   Stage 1: Initialize $\overline{\theta}_0 = \underline{\theta}_0 = 0$ and $\overline{\pi}_0(\cdot|s) = \underline{\pi}_0(\cdot|s) = Unif(\mathcal{A})$ for all $s \in \mathcal{S}$.

24 **for** $k = 0, 1, \cdots, K$ **do**

25    **for** $t = 0, 1, \cdots, T - 1$ **do**

26       Initialize $\overline{\theta}_{k,0} = \overline{\theta}_k, \underline{\theta}_{k,0} = \underline{\theta}_k$ and $\overline{\pi}_{k,0}(\cdot|s) = \overline{\pi}_k(\cdot|s), \underline{\pi}_{k,0}(\cdot|s) = \underline{\pi}_k(\cdot|s)$ for all $s \in \mathcal{S}$.

27       Sample *i.i.d.* $\{(s_i, a_i, b_i, s_i'), a_i', b_i'\}_{i \in [n]}$ with policy pair $\hat{\overline{\pi}}_{k,t}, \hat{\underline{\pi}}_{k,t}$, where $s_i' = f(s_i, a_i, b_i) + \varepsilon$.

28       Initialize $\hat{w}_{k,t,0} = 0$.

29       **for** $l = 0, 1, \cdots, L - 1$ **do**

30          Solve

$$\hat{w}_{k,t,l+1} = \arg\min_w \left\{ \sum_{i \in [n]} \left( \phi(s_i, a_i, b_i)^\top w - r(s_i, a_i, b_i) - \gamma \phi(s_i', a_i', b_i')^\top \hat{w}_{k,t,l} \right)^2 \right\} \tag{47}$$

31       Update $\overline{\theta}_{k,t+1} = \overline{\theta}_{k,t} + \eta \hat{w}_{k,t,L}, \underline{\theta}_{k,t+1} = \underline{\theta}_{k,t} - \eta \hat{w}_{k,t,L}$ and

$$\overline{\pi}_{k,t+1}(a|s) \propto \exp(\mathbb{E}_{b \sim \underline{\pi}_{k,t}(\cdot|s)}[\phi(s, a, b)]^\top \overline{\theta}_{k,t+1}), \tag{48}$$

$$\underline{\pi}_{k,t+1}(b|s) \propto \exp(\mathbb{E}_{a \sim \overline{\pi}_{k,t}(\cdot|s)}[\phi(s, a, b)]^\top \underline{\theta}_{k,t+1}). \tag{49}$$

32    Output $\underline{\pi}_k = \frac{1}{T} \sum_{t=1}^T \underline{\pi}_{k,t}$.

33    Stage 2: Initialize $\overline{\theta}'_{k,0} = 0$ and $\overline{\pi}'_{k,0}(\cdot|s) = Unif(\mathcal{A})$.

     **for** $t = 0, 1, \cdots, T - 1$ **do**

34       Sample *i.i.d.* $\{(s_i, a_i, b_i, s_i'), a_i', b_i'\}_{i \in [n]}$ with policy pair $\hat{\overline{\pi}}_k^t, \hat{\underline{\pi}}_k$, where $s_i' = f(s_i, a_i, b_i) + \varepsilon$.

35       Initialize $\hat{\zeta}_{k,t,0} = 0$.

36       **for** $l = 0, 1, \cdots, L - 1$ **do**

37          Solve

$$\hat{\zeta}_{k,t,l+1} = \arg\min_\zeta \left\{ \sum_{i \in [n]} \left( \phi(s_i, a_i, b_i)^\top \zeta - r(s_i, a_i, b_i) - \gamma \phi(s_i', a_i', b_i')^\top \hat{\zeta}_{k,t,l} \right)^2 \right\} \tag{50}$$

38       Update $\overline{\theta}'_{k,t+1} = \overline{\theta}'_{k,t} + \eta \hat{\zeta}_{k,t,L}$ and

$$\overline{\pi}'_{k,t+1}(a|s) \propto \exp(\mathbb{E}_{b \sim \underline{\pi}_k(\cdot|s)}[\phi(s, a, b)]^\top \overline{\theta}'_{k,t+1}). \tag{51}$$

39    Output $\overline{\pi}_k = \overline{\pi}'_{k,\hat{t}}$, where $\hat{t} \in [T]$ is selected uniformly at random.

---

$$\eta \langle \mathbb{E}_{a \sim \hat{\overline{\pi}}_{k,t}}[\hat{\hat{Q}}(s, a, \cdot)], \underline{\pi}(\cdot|s) - \underline{\pi}_{k,t+1}(\cdot|s) \rangle$$

$$= \eta \langle \mathbb{E}_{a \sim \hat{\overline{\pi}}_{k,t}}[\hat{\hat{Q}}(s, a, \cdot)], \underline{\pi}(\cdot|s) - \underline{\pi}_{k,t}(\cdot|s) \rangle$$

$$+ \eta \langle \mathbb{E}_{a \sim \hat{\overline{\pi}}_{k,t}}[\hat{\hat{Q}}(s, a, \cdot)], \underline{\pi}_{k,t}(\cdot|s) - \underline{\pi}_{k,t+1}(\cdot|s) \rangle$$

$$\leq \eta \langle \mathbb{E}_{a \sim \hat{\overline{\pi}}_{k,t}}[\hat{\hat{Q}}(s, a, \cdot)], \underline{\pi}(\cdot|s) - \underline{\pi}_{k,t}(\cdot|s) \rangle$$

$$+ \frac{\eta^2}{2(1 - \gamma)^2} + KL(\underline{\pi}_{k+1}(\cdot|s), \underline{\pi}_k(\cdot|s))$$

Using these estimations, we have that

$$\langle \mathbb{E}_{a\sim\hat{\bar{\pi}}_{k,t}}[\hat{Q}^{\hat{\bar{\pi}}_{k,t},\hat{\underline{\pi}}_{k,t}}(s,a,\cdot)], \underline{\pi}_{k,t}(\cdot|s) - \underline{\pi}(\cdot|s)\rangle + \frac{1}{\eta}KL(\underline{\pi}(\cdot|s), \underline{\pi}_{k,t+1}(\cdot|s))$$

$$\leq \frac{1}{\eta}KL(\underline{\pi}(\cdot|s), \underline{\pi}_{k,t}(\cdot|s)) + \frac{\eta}{2(1-\gamma)^2} + 2\eta\|\hat{Q}^{\hat{\bar{\pi}}_{k-1},\hat{\underline{\pi}}_{k-1}} - \hat{\hat{Q}}\|_{\nu_{\hat{\bar{\pi}}_{k-1},\hat{\underline{\pi}}_{k-1}}}$$

Summing the inequality, we get

$$\frac{1}{T}\sum_{t=1}^{T}\langle\mathbb{E}_{a\sim\hat{\bar{\pi}}_{k,t}}[\hat{\hat{Q}}^{\bar{\pi}_{k,t},\underline{\pi}_{k,t}}(s,a,\cdot)], \underline{\pi}_{k,t}(\cdot|s) - \underline{\pi}(\cdot|s)\rangle$$

$$\leq \frac{1}{\eta T}KL(\underline{\pi}, \underline{\pi}_{k,0}) + 2\eta\|\hat{Q}^{\hat{\bar{\pi}}_{k-1},\hat{\underline{\pi}}_{k-1}} - \hat{\hat{Q}}\|_{\nu_{\hat{\bar{\pi}}_{k-1},\hat{\underline{\pi}}_{k-1}}}$$

$$\leq \frac{1}{\eta T}\log\frac{1}{|\mathcal{A}|} + \frac{\eta}{2(1-\gamma)^2} + 2\eta\|\hat{Q}^{\hat{\bar{\pi}}_{k-1},\hat{\underline{\pi}}_{k-1}} - \hat{\hat{Q}}\|_{\nu_{\hat{\bar{\pi}}_{k-1},\hat{\underline{\pi}}_{k-1}}}$$

Thus

$$\mathbb{E}\left[\max_{\bar{\pi}}\bar{\pi}Q_{k-1}\underline{\pi}_k - \min_{\underline{\pi}}\bar{\pi}_kQ_{k-1}\underline{\pi}\right]$$

$$\leq \frac{\eta}{2(1-\gamma)^2} + \frac{32\gamma\beta}{(1-\gamma)^2} + \frac{1}{\eta T}\log\frac{1}{|\mathcal{A}|} + 2\eta\|\hat{Q}^{\hat{\bar{\pi}}_{k-1},\hat{\underline{\pi}}_{k-1}} - \hat{\hat{Q}}\|_{\nu_{\hat{\bar{\pi}}_{k-1},\hat{\underline{\pi}}_{k-1}}}$$

$\square$

**Lemma E.9.** *In Stage 2 of Algorithm 4,*

$$\mathbb{E}\frac{1}{T}\sum_{t=0}^{T}V^{\bar{\pi}_k^*,\underline{\pi}_k}(s_0) - V^{\bar{\pi}'_{k,t},\underline{\pi}_k}(s_0)$$

$$\leq O\left(\frac{\log|\mathcal{A}|}{(1-\gamma)\eta T} + \frac{\eta}{(1-\gamma)^3} + \frac{\gamma\beta}{\eta(1-\gamma)^3} + \frac{\|\hat{Q}^{\bar{\pi}'_{k,t},\underline{\pi}_k} - \hat{\hat{Q}}\|_{\nu_{\bar{\pi}'_{k,t},\underline{\pi}_k}}}{1-\gamma}\right).$$

*Proof.* Denote $\hat{\hat{Q}} = Q^{\hat{\bar{\pi}}'_{k,t},\hat{\underline{\pi}}_k}$. By the update rule of the algorithm,

$$KL(\bar{\pi}(\cdot|s), \bar{\pi}'_{k,t+1}(\cdot|s)) \leq KL(\bar{\pi}(\cdot|s), \bar{\pi}'_{k,t}(\cdot|s))$$

$$- \eta\langle\mathbb{E}_{b\sim\hat{\underline{\pi}}_k}[\hat{Q}^{\hat{\bar{\pi}}'_{k,t},\hat{\underline{\pi}}_k}(s,\cdot,b)], \bar{\pi}(\cdot|s) - \bar{\pi}'_{k,t+1}(\cdot|s)\rangle - KL(\bar{\pi}'_{k,t+1}(\cdot|s), \bar{\pi}'_{k,t}(\cdot|s))$$

$$= KL(\bar{\pi}(\cdot|s), \bar{\pi}'_{k,t+1}(\cdot|s)) - KL(\bar{\pi}'_{k,t+1}(\cdot|s), \bar{\pi}'_{k,t}(\cdot|s))$$

$$- \eta\langle\mathbb{E}_{b\sim\hat{\underline{\pi}}_k}[\hat{Q}^{\hat{\bar{\pi}}'_{k,t},\hat{\underline{\pi}}_k}(s,\cdot,b) - \hat{\hat{Q}}(s,\cdot,b)], \bar{\pi}(\cdot|s) - \bar{\pi}'_{k,t+1}(\cdot|s)\rangle$$

$$- \eta\langle\mathbb{E}_{b\sim\hat{\underline{\pi}}_k}[\hat{\hat{Q}}(s,\cdot,b)], \bar{\pi}'_{k,t}(\cdot|s) - \bar{\pi}'_{k,t+1}(\cdot|s)\rangle$$

$$- \eta\langle\mathbb{E}_{b\sim\hat{\underline{\pi}}_k}[Q^{\hat{\bar{\pi}}'_{k,t},\hat{\underline{\pi}}_k}(s,\cdot,b)], \bar{\pi}(\cdot|s) - \bar{\pi}'_{k,t}(\cdot|s)\rangle$$

$$- \eta\langle\mathbb{E}_{b\sim\hat{\underline{\pi}}_k}[\hat{\hat{Q}}(s,\cdot,b) - Q^{\hat{\bar{\pi}}'_{k,t},\hat{\underline{\pi}}_k}(s,\cdot,b)], \bar{\pi}(\cdot|s) - \bar{\pi}'_{k,t}(\cdot|s)\rangle$$

We bound the inner products similarly

$$\eta\langle\mathbb{E}_{b\sim\hat{\underline{\pi}}_k}[\hat{Q}^{\hat{\bar{\pi}}'_{k,t},\hat{\underline{\pi}}_k}(s,\cdot,b) - \hat{\hat{Q}}(s,\cdot,b)], \bar{\pi}(\cdot|s) - \bar{\pi}'_{k,t+1}(\cdot|s)\rangle \leq O(\eta\|\hat{Q}^{\hat{\bar{\pi}}'_{k,t},\hat{\underline{\pi}}_k} - \hat{\hat{Q}}\|_{\nu_{\hat{\bar{\pi}}'_{k,t},\hat{\underline{\pi}}_k}})$$

$$\eta\langle\mathbb{E}_{b\sim\hat{\underline{\pi}}_k}[\hat{\hat{Q}}(s,\cdot,b)], \bar{\pi}'_{k,t}(\cdot|s) - \bar{\pi}'_{k,t+1}(\cdot|s)\rangle \leq \frac{\eta^2}{2(1-\gamma)^2} + KL(\bar{\pi}'_{k,t+1}(\cdot|s), \bar{\pi}'_{k,t}(\cdot|s))$$

and

$$\eta\langle\mathbb{E}_{b\sim\underline{\pi}_k}[\hat{\hat{Q}}(s,\cdot,b) - Q^{\bar{\pi}'_{k,t},\underline{\pi}_k}(s,\cdot,b)], \bar{\pi}(\cdot|s) - \bar{\pi}'_{k,t}(\cdot|s)\rangle \leq \frac{16\gamma\beta}{(1-\gamma)^2}$$

**Algorithm 5** Spectral Dynamic Embedding Policy Optimization with Neural Networks

**Data:** Transition Model $s' = f(s, a, b) + \varepsilon$ where $\varepsilon \sim \mathcal{N}(0, \sigma^2 I_d)$, Reward Function $r(s, a, b)$, Number of Random/Nyström Feature $m$, Number of Nyström Samples $n_{\mathrm{Nys}} \geq m$, Nyström Sampling Distribution $\mu_{\mathrm{Nys}}$, Factorization Scale $\alpha$, Learning Rate $\eta_{actor}$ and $\eta_{critic}$

**Result:** $\overline{\pi}_{\overline{\theta}}, \underline{\pi}_{\underline{\theta}}$

1 Generate $\phi(s, a, b)$ using Algorithm 1 or Algorithm 2.

2 Initialize $\overline{\pi}_{\overline{\theta}}, \underline{\pi}_{\underline{\theta}}, \overline{Q}_{\overline{\psi}}, \underline{Q}_{\underline{\psi}}$.

3 **for** *each iteration* **do**

4    **for** *each environment step* **do**

5       Sample $\{(s_i, a_i, b_i, s_i', a_i', b_i')\}$ to replay buffer.

6    **for** *each gradient step* **do**

7       Update $\overline{\theta}, \underline{\theta}$ by minimizing eqs. (55) and (57), respectively.

8       Update $\overline{\psi}, \underline{\psi}$ by minimizing eqs. (61) and (62), respectively.

---

Consequently, we have that

$$\langle \mathbb{E}_{b \sim \underline{\pi}_k}[Q^{\overline{\pi}'_{k,t}, \underline{\pi}_k}(s, \cdot, b)], \overline{\pi}(\cdot|s) - \overline{\pi}'_{k,t}(\cdot|s) \rangle + \frac{1}{\eta} KL(\overline{\pi}(\cdot|s), \overline{\pi}'_{k,t+1}(\cdot|s))| \tag{52}$$

$$\leq \frac{1}{\eta} KL(\overline{\pi}(\cdot|), \overline{\pi}'_{k,t}(\cdot|s)) + O(\frac{\gamma\beta}{(1-\gamma)^2} + \eta(\frac{1}{(1-\gamma)^2} + \|\hat{Q}^{\overline{\pi}'_{k,t}, \underline{\pi}_k} - \hat{\hat{Q}}\|_{\nu_{\overline{\pi}'_{k,t}, \underline{\pi}_k}})) \tag{53}$$

By the performance difference lemma

$$V^{\overline{\pi}^*_k, \underline{\pi}_k}(s_0) - V^{\overline{\pi}'_{k,t}, \underline{\pi}_k}(s_0) = \frac{1}{1-\gamma} \mathbb{E}_{s \sim d_{s_0}^{\overline{\pi}^*_k, \underline{\pi}_k}} \langle \mathbb{E}_{b \sim \underline{\pi}_k(\cdot|s)} Q^{\overline{\pi}'_{k,t}, \underline{\pi}_k}(s, \cdot, b), \overline{\pi}^*_k(\cdot|s) - \overline{\pi}'_{k,t}(\cdot|s) \rangle.$$

Summing the inequality, we get

$$\mathbb{E} \frac{1}{T} \sum_{t=0}^{T} V^{\overline{\pi}^*_k, \underline{\pi}_k}(s_0) - V^{\overline{\pi}'_{k,t}, \underline{\pi}_k}(s_0)$$

$$\leq O(\frac{\log|\mathcal{A}|}{(1-\gamma)\eta T} + \frac{\eta}{(1-\gamma)^3} + \frac{\gamma\beta}{\eta(1-\gamma)^3} + \frac{\|\hat{Q}^{\overline{\pi}'_{k,t}, \underline{\pi}_k} - \hat{\hat{Q}}\|_{\nu_{\overline{\pi}'_{k,t}, \underline{\pi}_k}}}{1-\gamma}) \tag{54}$$

$\square$

**Proposition E.5.** *For the output of player2,*

$$E_{s \sim \mu_0}\left[\max_{\overline{\pi}} V^{\overline{\pi}, \underline{\pi}_K}(s) - V^*(s)\right]$$

$$\leq O\left(\frac{CK}{(1-\gamma)^2}(\frac{\log\frac{1}{\underline{c}}}{\eta T} + \frac{\eta}{(1-\gamma)^2} + \frac{\gamma\beta}{\eta(1-\gamma)^2} + \eta\|\hat{Q}^{\overline{\pi}'_{k,t}, \underline{\pi}_k} - Q^{\overline{\pi}'_{k,t}, \underline{\pi}_k}\|_{\nu_{\overline{\pi}'_{k,t}, \underline{\pi}_k}}\right.$$

$$\left. + \eta\|\hat{Q}^{\hat{\overline{\pi}}_{k-1}, \hat{\underline{\pi}}_{k-1}} - \hat{\hat{Q}}\|_{\nu_{\hat{\overline{\pi}}_{k-1}, \hat{\underline{\pi}}_{k-1}}}) + \frac{\gamma^K}{1-\gamma}\right).$$

*where $C$ is a problem-dependent constant.*

*Proof.* Combining Lemma E.8, Lemma E.9 and (44) gives the result. $\square$

## F SDEPO with Neural Networks

To deal with continuous action space, we propose a practical variant of `SDEPO`, named `SDEPO-NN`, to deal with continuous (or large-scale discrete ) action space. `SDEPO-NN` utilizes neural networks in policy $\pi$ and the state-action value function approximation $Q$, detailed in Algorithm F.

Specifically, we parameterize $\overline{\pi}, \underline{\pi}$ with $\overline{\psi}, \underline{\psi}$ and parameterize $\overline{Q}, \underline{Q}$ with $\overline{\theta}, \underline{\theta}$, respectively. Based on the spectral dynamic embedding proposed in our paper, we parameterize the $\overline{Q}$ function as $\overline{Q}_{\overline{\theta}}(s, a, b) = r(s, a, b) + \phi(s, a, b)^\top \overline{\theta}$ and parameterize the $\underline{Q}$ function as $\underline{Q}_{\underline{\theta}}(s, a, b) = r(s, a, b) + \phi(s, a, b)^\top \underline{\theta}$.

The $\overline{Q}$ function is trained to minimize the soft Bellman residual

$$J(\overline{\theta}) = \mathbb{E}_{(s_t, a_t, b_t, s'_t, a'_t, b'_t) \sim \mathcal{D}} \left[ \frac{1}{2} (\overline{Q}_{\overline{\theta}}(s_t, a_t, b_t) - \hat{\overline{Q}}_{\overline{\theta}}(s_t, a_t, b_t))^2 \right], \tag{55}$$

with

$$\hat{\overline{Q}}_{\overline{\theta}}(s_t, a_t, b_t) = r(s_t, a_t, b_t) + \gamma[\overline{Q}_{\overline{\theta}}(s'_t, a'_t, b'_t) - \alpha \log \overline{\pi}_{\overline{\psi}}(a'_t | s'_t)]. \tag{56}$$

Similarly, The $\underline{Q}$ function is trained to minimize the soft Bellman residual

$$J(\underline{\theta}) = \mathbb{E}_{(s_t, a_t, b_t, s'_t, a'_t, b'_t) \sim \mathcal{D}} \left[ \frac{1}{2} (\underline{Q}_{\underline{\theta}}(s_t, a_t, b_t) - \hat{\underline{Q}}_{\underline{\theta}}(s_t, a_t, b_t))^2 \right], \tag{57}$$

with

$$\hat{\underline{Q}}_{\underline{\theta}}(s_t, a_t, b_t) = r(s_t, a_t, b_t) + \gamma[\underline{Q}_{\underline{\theta}}(s'_t, a'_t, b'_t) - \alpha \log \underline{\pi}_{\underline{\psi}}(b'_t | s'_t)]. \tag{58}$$

We restrict the policy to Gaussians and the policy parameters can be learned by minimizing

$$J(\overline{\psi}) = \mathbb{E}_{s_t \sim \mathcal{D}} \left[ KL \left( \overline{\pi}_{\overline{\psi}}(\cdot | s_t) \| \frac{\exp(\mathbb{E}_{b_t \sim \underline{\pi}_{\underline{\psi}}}[Q_\theta(s_t, \cdot, b_t)])}{Z_{\theta,1}(s_t)} \right) \right] \tag{59}$$

and

$$J(\underline{\psi}) = \mathbb{E}_{s_t \sim \mathcal{D}} \left[ KL \left( \underline{\pi}_{\underline{\psi}}(\cdot | s_t) \| \frac{\exp(\mathbb{E}_{a_t \sim \overline{\pi}_{\overline{\psi}}}[Q_\theta(s_t, a_t, \cdot)])}{Z_{\theta,2}(s_t)} \right) \right], \tag{60}$$

where $\mathcal{D}$ is the replay buffer, $Z_{\theta,1}(s_t)$ and $Z_{\theta,2}(s_t)$ are normalization factors for the distributions.

To lower variance, we reparameterize the policy using a neural network transformation $a_t = f_{\overline{\psi}}(\epsilon_t; s_t)$ and $b_t = f_{\underline{\psi}}(\epsilon'_t; s_t)$ where $\epsilon_t$ and $\epsilon'_t$ are input noise vectors, sampled from a Gaussian. We can now rewrite the objectives in Equations 59 and 60 as

$$J(\overline{\psi}) = \mathbb{E}_{s_t \sim \mathcal{D}, \epsilon_t, \epsilon'_t \sim \mathcal{N}} \left[ \log \overline{\pi}_{\overline{\psi}}(f_{\overline{\psi}}(\epsilon_t; s_t) | s_t) - Q_\theta(s_t, \overline{\pi}_{\overline{\psi}}(f_{\overline{\psi}}(\epsilon_t; s_t) | s_t), \underline{\pi}_{\underline{\psi}}(f_{\underline{\psi}}(\epsilon'_t; s_t) | s_t)) \right]$$
$$\tag{61}$$

and

$$J(\underline{\psi}) = \mathbb{E}_{s_t \sim \mathcal{D}, \epsilon_t, \epsilon'_t \sim \mathcal{N}} \left[ \log \underline{\pi}_{\underline{\psi}}(f_{\underline{\psi}}(\epsilon'_t; s_t) | s_t) - Q_\theta(s_t, \overline{\pi}_{\overline{\psi}}(f_{\overline{\psi}}(\epsilon_t; s_t) | s_t), \underline{\pi}_{\underline{\psi}}(f_{\underline{\psi}}(\epsilon'_t; s_t) | s_t)) \right]$$
$$\tag{62}$$

where $\overline{\pi}_{\overline{\psi}}$ and $\underline{\pi}_{\underline{\psi}}$ are defined implicitly in terms of $f_{\overline{\psi}}$ and $f_{\underline{\psi}}$, respectively.

