# OpenReview forum: "Solving Zero-Sum Markov Games with Continuous State via Spectral Dynamic Embedding"
_NeurIPS.cc/2024/Conference — NeurIPS 2024 poster_

### Official Review · Reviewer_DHon · 2024-07-05

**Soundness:** 2
**Presentation:** 2
**Contribution:** 3
**Rating:** 7
**Confidence:** 4

**Summary:**

This paper studies the two-player zero-sum stochastic Markov games (2p0s-MGs) with large scale or continuous state spaces. These problems have a large cardinality and function approximation methods are needed. The paper consider a spectral dynamic embedding method and proposed SDEPO. This methods utilized the transition dynamics in the construction of the state-space value function. SDEPO is able to converge with order $1/\epsilon$, which matches the optimal rate in single agent RL.
Theorems are provided for the last iterate convergence of the SDEPO algorithm. The effectiveness of the algorithm has been verified in games against baseline methods.

Generally the paper is well structured, but section 3-4 should be better explained while section 5 and 6 focused on the "practical algorithms" is stretched quite far from the analytical results in the previous sections.

**Strengths:**

The function approximation approach for markov games is a necessity for those problems with large/infinite state cardinality. It is indeed true that the dynamics of the problem was not utilized in previous methods. This algorithm seems to be the first work addressing this.

This work adapted the spectral dynamic embedding for stochastic nonlinear control problems and proposed SDEPO, the motivation is clear and well stated.

**Weaknesses:**

The only evaluation for the proposed algorithm is a rate of success in playing games with baseline algorithms. To the reviewer, this seems to be very limited. For a submission with strong theoretical focus, current result fails to validate the convergence properties of the proposed algorithm.

The sections for main results are not very well-written and is a bit difficult to read, more explanation would be appreciated. Although this could be due to the page limit.

Assumption 5 is in fact quite strong, a brief discussion on the impact and reasoning should be provided.

Section 5 and 6 seems a bit rushed and is intended to bring out the neural networks, the prior sections discussed the setting with tabular actions, where in these sections the action space is seen as continuous and more algorithms have been added, with no analytical results. I suggest the authors focusing on the existing setting with better presentation, explanation and more experiments.

Another problem this paper did not address is what are the current existing algorithms involving dynamics and function approximation in the single agent setting. The single agent RL with function approximation literature should be somewhat addressed in general.

**Questions:**

What is the effect of truncation w.r.t. the later regularity condition assumptions? Does a more limiting truncation negatively impact these assumptions on the problem?

What is the reason for the consideration of one-sided $\epsilon$−optimal Nash equilibrium? The author stated that many existing works also consider this, but an explanation would be appreciated.

**Limitations:**

The authors did not fully address the limitations of this paper. The authors mentioned that the algorithm is non-independent and relies on some assumptions. There are some other limitations of this work, which has been raised in the previous sections.

---

> ### Author Rebuttal · Authors · 2024-08-07
>
> We are grateful for your positive feedback on our manuscript. We give the point-to-point responses to the weaknesses and questions as follows.
>
> [Empirical evaluation:]
>
> We add a simulated experiment to validate the effectiveness of SDEPO directly (please see the Global Response for further details). Our original empirical study mainly focus on SDEPO with a neural network policy, and show its ability to deal with complex tasks with continuous action space.
>
> [Main result:]
>
> Due to space limit, we only presented the main results in Section 4 and provided detailed explanations in the appendix. Specifically, we thoroughly discussed the sources of errors in policy evaluation and compared the two feature generation methods in Appendix D, and analyzed the results of policy optimization in Appendix E. We will add more explanations in the main context in our revision.
>
> [Assumption 5:]
>
> This assumption is commonly used in the literature [1,2] and can be archived with $\epsilon$-greedy policy. Actually, we focus on finding stochastic policies converging to the Nash equilibrium (NE) of Markov games (MGs). While a deterministic policy NE may not exist for Markov games, a stochastic policy NE always exists [3]. A classic example is the rock-paper-scissors game, where the only NE is achieved by stochastic policies mixing between the three actions equally. Hence, we consider stochastic policy here and assume $\pi(a|s)\geq \underline c$, which can be ensured this with $\epsilon-$greedy policy and set $\underline{c}$ to be $\epsilon$.
>
> [Section 5 and 6:]
>
> In section 5, we extend SDEPO with neural networks policy. This extension is designed to accommodate continuous action space for complex tasks, where SDEPO could not handle. To bring the interests in a broader audience of computer science literature, we believe it is valuable to derive practical implementations from the theoretical-guaranteed basis. Additionally, we conduct a simulated experiment to validate the effectiveness of SDEPO directly (please check Global Response).
>
> [Related work in the single agent setting:]
>
> Function approximation in single-agent RL has been extensively studied in recent years to achieve a better sample complexity that depends on the complexity of function approximators rather than the size of the state/action space. One line of work studies RL with linear function approximation [4,5]. Typically, these methods assume the optimal value function can be well approximated by linear functions, and achieve polynomial sample efficiency guarantees related to feature dimension under certain regularity conditions. Another line of works studied the MDPs with general nonlinear function approximations [6,7]. [6,7] present algorithms with PAC guarantees for problems with low Bellman rank and low BE dimension, respectively. We note that MGs are inherently more complex than MDPs due to their min-max nature and it is generally difficult to directly extend these results to the dual-player dynamic setting of MGs.
>
> [The effect of truncation w.r.t. the later regularity conditions:]
>
> In our paper, we use the Random Features/Nystrom Features generation methods to provide efficient truncated feature embeddings to represent the transition kernel $P$ and the reward function $r$. Actually, we prove, in Theorem 4,  that the evaluation error of Q-function with $m$ truncated Random /Nyström Features is $O(m^{-1/2}+m^3/{\Upsilon^2_1\Upsilon_2n^{1/2}})$/ $O(m^{-1}+m^3/{\Upsilon^2_1 \Upsilon_2 n^{1/2}})$, where $n$ is the per-iteration sample size and $\Upsilon_1$ and $\Upsilon_2$ denotes regularity constants relating the eigen-spectrum of the stationary distribution (in Assumption 4). Hence, to achieve a small policy evaluation error, Theorem4 indicates that the less regular in the eigen-spectrum of the stationary distribution (with smaller $\Upsilon_1$ and $\Upsilon_2$), the more features and samples is needed. Typically, larger truncations provides a better approximation of the original infinite-dimensional function space, but at the cost of increased sample complexity. Thus, it is essential to strike a balance between approximation error and statistical error when selecting the optimal truncation size. Larger $\Upsilon_1$ and $\Upsilon_2$ allow us to select a smaller truncation size and thereby enhance the precision of the predicted Q-function and convergence to NE.
>
> [The reason for the consideration of one-sided NE:]
>
> One-side NE is a common objective in the MG literature [1,8], as it can be directly extended to establish a two-sided NE. Specifically, we can apply algorithms with the roles switched, and add the two one-sided bound to achieve a two-sided NE[8].
>
> [1] Ahmet Alacaoglu, et al. A natural actor-critic framework for zero-sum markov games. In International Conference on Machine Learning, pages 307–366. PMLR, 2022
>
> [2] Lan, G. Policy mirror descent for reinforcement learning: Linear convergence, new sampling complexity, and generalized problem classes. arXiv:2102.00135, 2021
>
> [3] John Nash. Non-cooperative games. Annals of mathematics, pp. 286–295, 1951.
>
> [4] Lin Yang and Mengdi Wang. Sample-optimal parametric q-learning using linearly additive features. In International Conference on Machine Learning, pages 6995–7004. PMLR, 2019.
>
> [5] Chi Jin, et al. Provably efficient reinforcement learning with linear function approximation. In Conference on Learning Theory, pages 2137–2143. PMLR, 2020.
>
> [6] Nan Jiang, et al. Contextual decision processes with low bellman rank are pac-learnable. In International Conference on Machine Learning, pages 1704–1713. PMLR, 2017.
>
> [7] Chi Jin, et al. Bellman eluder dimension: New rich classes of rl problems, and sample-efficient algorithms. Advances in Neural Information Processing Systems, 34, 2021.
>
> [8] Yulai Zhao, et al. Provably efficient policy optimization for two-player zero-sum markov games. In International Conference on Artificial Intelligence and Statistics, pages 2736–2761. PMLR, 2022

---

> > ### Comment · Reviewer_DHon · 2024-08-07
> >
> > Thank you for the detailed explanations provided by the authors and the newly updated numerical results. The authors have addressed my questions and I have decided to increase my score.

---

> > > ### Author Response · Authors · 2024-08-14
> > >
> > > We sincerely thank you for recognizing our efforts in rebuttal. We greatly appreciate your decision to increase the score. We will incorporate the discussion on the theoretical analyses and newly updated numerical results in our revision.

---

### Official Review · Reviewer_sWhd · 2024-07-08

**Soundness:** 3
**Presentation:** 3
**Contribution:** 3
**Rating:** 4
**Confidence:** 3

**Summary:**

This paper proposes a new algorithm named Spectral Dynamic Embedding Policy Optimization (SDEPO) to solve the zero-sum Markov games with continous state and finite actions. The convergence analysis indicates that the proposed method achieves the best-known sample complexity as the case of finite-state space; this paper is the first theoretical result in handling the continuous state space with known dynamic and infinite horizon.

**Strengths:**

This paper is the first result for solving the NE of infinite-horizon two-player zero-sum Markov games  with continuous state space when the dynamic is known. Moreover, this paper resents sufficient introduction on the technical backgrounds and preliminaries. All assumptions are clearly listed. Lastly, the theoretical results are verified using emprical experiments.

**Weaknesses:**

1. The Assumption 1 is not reasonable. It says, whatever the state $s$ and the action $(a,b)$ are, the agent can move to any state $s'$ with a positive probability. Please correct me if I am wrong.

2. I am confused about what is new in the Spectral Dynamic Embedding method. It seems that both Bochner and Mercer theorem are well-known. This paper simply applies them to represent the transition probability and the reward using some kernels. Then everything is the same as traditional method in RL.

3. A mild comment on Assumption 3: Since the optimal policy might be deterministic, it means that $\pi(a|s)$ is likely to be zero for some $a$. During the training, the policy $\pi_k$ will tend to the optimal policy; the mass at non-optimal action will also approach to $0$. It means if  $\underline{c}$ is larger than $\epsilon$, then $\pi_k$ will never converge to the optimal action in the sense of $L_\infty$ norm. From my understanding, the author needs to set $\underline{c}$ to be $\epsilon$ and it won't affect the complexity.

**Questions:**

Q1:  Can the author justify the use of Assumption 1? It seems to be unrealistic in RL.

Q2: This paper seems to be a simple combination of linear MDP + policy gradient. What is the novel part of this work? I feel hard to consider representing $\mathbb{P}$ and $r$ using kernel methods as a new thing.

---

> ### Author Rebuttal · Authors · 2024-08-07
>
> We are grateful for the effort you have dedicated to our paper, and we give the one-to-one responses to the weakness and questions as follows.
>
> Response to W1&Q1:
>
> In Assumption 1, we assume that the transition function satisfies that $s_{t+1}=f(s_t,a_t,b_t)+\epsilon_t$, which means that next state is determined by certain deterministic function $f$ with an additional random noise $\epsilon_t$. Actually, such transition function exists in many RL tasks, such as robot control [3] and real-time bidding games [4], where the transition is almost determined by the actions with unpredictable system factors (internal system chaos and external disturbances). Typically, the random noise is often assumed as Gaussian and Assumption 1 is common in the literature [1,2,3,4].
>
> Even for the Gaussian noise, most of the transition probability would concentrate in a limited area according to the 3-$\sigma$ rule and $s_{t+1}$ would most likely to be near $f(s_t,a_t,b_t)$, though “the agent can move to any state with a positive probability”. Moreover, our result can be generalized to compactly supported noises, e.g. the truncated Gaussian noise. We will add more discussion about this in our revision.
>
> Response to W2&Q2:
>
> The reviewer doubts about the novelty of the Spectral Dynamic Embedding method and, furthermore, our proposed SDEPO method. Here, we note that our goal is to propose a provably efficient method for two-player zero-sum Markov games (MGs) with a continuous state space. The process of SDEPO seems straightforward, i.e., it first generates $m$ feature embeddings with Spectral Dynamic Embedding, and then applies natural gradient policy improvements for each player based on generated feature embeddings. Note that such feature embedding generation + policy optimization structure is quite common in the RL literature [5,6]. The difficulty lies in choose proper feature embedding/policy optimization methods, and prove that the two parts can interact well to achieve the convergence to the Nash Equilibrium (NE).
>
> Basically, a desirable feature embedding method should provide a good approximation to the underlying dynamics of MG with a modest $m$. That is exactly what our two specific feature generation methods achieve. The Random/Nystrom Features methods could provide efficient finite linear approximations to represent the transition kernel $P$, as finite-dimensional truncations of the Bochner decomposition and Mercer decomposition, respectively.
>
> In the policy optimization procedure, we choose the natural policy gradient method to adjust the policy of each player iteratively in an alternating fashion. Specifically, for each player, it first conducts least square policy evaluation to estimate the state-action Q value function of current policy upon based on the generated features and then improve the policy by natural policy gradient based on the estimated Q function.
>
> The critical part lies in the theoretical analyses. First, in Theorem 1, we prove that the policy evaluation error of each player’s Q-function is $O(m^{-1/2}+m^3/n^{1/2})$/$O(m^{-1}+m^3/n^{1/2})$ ,where $n$ is the per-iteration sample size. With $m$ being $O(n^{1/7})$/$O(n^{1/8})$ for Random/Nystrom features,  an optimal evaluation error can be achieved. More importantly, we bound the error of the one-side Nash equilibrium explicitly with the policy evaluation error in Proposition 5, and establish the overall iteration complexity to achieve an $\epsilon−$optimal NE.
>
> Though Spectral Dynamic Embedding and the natural gradient policy strategy seem common, the merit of our method lies in identifying effective embeddings to enhance policy optimization and analyze the impact of evaluation error on the convergence to NE.
>
> Besides, we note that we focus on MGs instead of MDPs, which are inherently more complex due to their min-max nature of MGs. Unlike methods for MDPs that leverage a single-player perspective, our approach must address the dual-player dynamics of MGs. Furthermore, our problem further differentiates linear MG. Unlike algorithms dealing with linear MG [7,8], where the underlying dynamics is assumed to be linear and known, we only assume that Assumption 1 holds and construct finite-dimensional dimension embeddings.
>
> Response to W3:
>
> In this paper, we focus on finding policies converging to the NE of MGs. Note that a deterministic policy NE may not exist for a general Markov game while a stochastic policy NE always exists for games with finite actions [9]. A classic example is the rock-paper-scissors game, where the only NE is achieved by stochastic  policies mixing between the three actions equally. This is different from the single-agent MDPs, where a deterministic optimal policy always exists. Hence, we consider stochastic policy here and assume $\pi(a|s)\geq \underline c$. We can ensure this with $\epsilon-$greedy policy and set $\underline{c}$ to be $\epsilon$.
>
> [1] Ren, T., et. al. A free lunch from the noise: Provable and practical exploration for representation learning. UAI 2022.
>
> [2] Horia, M.,et. al. Certainty equivalence is efficient for linear quadratic control. NIPS 2019.
>
> [3] Ignasi, C., et. al. Model-based reinforcement learning via meta-policy optimization. CoRL 2018.
>
> [4] Hambly, B., et. al. Policy gradient methods find the Nash equilibrium in N-player general-sum linear-quadratic games. arXiv:2107.13090, 2021.
>
> [5] Chengzhuo, N., et. al. Representation learning for general-sum low-rank markov games. arXiv:2210.16976.
>
> [6] Masatoshi, U., et. al. Representation learning for online and offline rl in low-rank mdps. arXiv:2110.04652.
>
> [7] Zixiang C., et. al. Almost optimal algorithms for two-player markov games with linear function approximation. arXiv:2102.07404.
>
> [8] Qiaomin X., et. al.. Learning zero-sum simultaneous-move markov games using function approximation and correlated equilibrium. In COLT 2020.
>
> [9] John Nash. Non-cooperative games. Annals of mathematics, pp. 286–295, 1951.

---

> > ### Author Response · Authors · 2024-08-14
> >
> > Dear Reviewer [sWhd],
> >
> > Thank you again for your constructive feedback. The end of the discussion period is close. We would be grateful if we could hear your feedback regarding our answers to the reviews. Here, we clarify the following part of our work again.
> >
> > [Novelty of this paper] The primary goal of our paper is to propose a provably efficient method for two-player zero-sum stochastic Markov games with a continuous state space. Achieving Nash equilibria of Markov games is inherently more complex than identifying optimal policies of MDPs due to the min-max nature of Markov games and the non-stationary environment that each player faces. Our method first generates the $m$ feature embeddings through Spectral Dynamic Embedding method, and then evaluates the Q functions based on the embeddings and applies natural gradient policy improvements for each player based on the evaluated Q functions. Unlike linear Markov games, where the underlying representation is assumed to be known, we need to choose a proper feature embedding method and a suitable policy optimization strategy. We prove that the two parts can interact well to achieve the convergence of policies to a Nash equilibrium with a desirable overall iteration complexity. Specifically, we theoretically analyze the policy evaluation error of each player’s Q-function based on the embeddings and bound the error of the one-side Nash equilibrium explicitly with the policy evaluation error. Note that even the kernel method and natural gradient method seems common in the RL literature, the merit of our method lies in identifying effective embeddings to enhance policy optimization and analyze the impact of evaluation error on the convergence to NE.
> >
> > [The rationality of Assumption 1] Assumption 1 assumes that next state is determined by certain deterministic function with an additional Gaussian noise. Such assumption is actually realistic (satisfied in many RL tasks) and is commonly assumed in both theory and practice in the RL literature. Your main concern of assumption 1 is that it requires the transition probability to remain consistently positive, attributed to the Gaussian noise. Nevertheless, due to the $3-\sigma$ rule, most of next state’s probability mass would concentrate within a limited area. Moreover, our results can be generalized to compactly supported transition randomness which admits certain kernelized representation (such as truncated Gaussian). We will add more discussion about this in our revision.
> >
> >
> > We sincerely hope to engage in further discussions with you. Thank you for your time and consideration!

---

### Official Review · Reviewer_SoJH · 2024-07-14

**Soundness:** 3
**Presentation:** 3
**Contribution:** 3
**Rating:** 7
**Confidence:** 2

**Summary:**

The authors introduce an innovative approach to solving 2p0s-MGs with continuous state spaces, providing both theoretical guarantees and practical improvements over existing methods. The SDEPO algorithm and its variants offer efficient and scalable solutions for complex Markov games, potentially applicable to various domains in reinforcement learning.

**Strengths:**

1. This paper proposes a new Spectral Dynamic Embedding Policy Optimization algorithm that effectively addresses two-player zero-sum Markov games with continuous state space and finite action space.
2. To handle the finite action spaces, a practical variant of SDEPO is proposed to manage continuous action spaces, with empirical results showcasing its superior performance.
3. The complexity result of SDEPO matches the best-known results for policy gradient algorithms in the single-agent setting, proving its efficiency.

**Weaknesses:**

1. The spectral embedding methods can be computationally intensive in practice due to the complexity of handling spectral dynamic embeddings.
2. Why were these specific feature generation methods chosen? Is the proposed method sensitive to feature generation methods?
3. The experiments are somewhat limited, expanding the empirical section to include more complex and diverse scenarios would significantly strengthen the paper.

**Questions:**

Please see the weakness part.

**Limitations:**

Yes.

---

> ### Author Rebuttal · Authors · 2024-08-07
>
> We are grateful for your positive feedback on our manuscript. We give the point-to-point responses to the weaknesses and questions as follows.
>
> Response to Weakness 1:
>
> It is really a good question to consider the computational overhead of the spectral dynamic embeddings. Actually, in our SDEPO algorithm, the spectral embedding method would only be executed once at the beginning to generate $m$ embeddings. To achieve the SOTA $\tilde O(\frac{1}{(1-\gamma)^3 \epsilon})$ iteration complexity, $m$ should be $\tilde O(n^{1/7})$/$\tilde O(n^{1/8})$ for random features/Nystrom features and the overall computational cost of random features/Nystrom feature generation procedure is $\tilde O(n^{1/7})$/$\tilde O(n^{3/8})$, where $n$ is the per-iteration sample size in the policy optimization procedure of SDEPO. Hence, the computational cost of the spectral embedding methods is mild compared to the policy optimization procedure in SDEPO. Actually, this phenomenon also exists in other represent learning method for Markov games, e.g., [1].
>
> Response to Weakness 2:
>
> Actually, the feature generation method lies at the core of our proposed SDEPO, i.e., SDEPO first generates $m$ feature embeddings and then conducts policy optimization based on the generated embeddings. A desirable feature embedding method should be able to provide a good approximation to the underlying dynamics of the Markov game with a modest $m$. That is exactly why the two specific feature generation methods achieve here. The Random Features/Nystrom Features generation methods could provide efficient approximations to represent the transition kernel $P$ and the reward function $r$, based on two well-known decompositions Bochner decomposition and Mercer decomposition, respectively. Actually, we prove that the evaluation error of each player’s Q-function with Random Features/Nyström Features is $O(m^{-1/2}+m^3/n^{1/2})$/$O(m^{-1}+m^3/n^{1/2})$ where $n$ is the per-iteration sample size in the policy optimization procedure of SDEPO. With $m$ being $\tilde O(n^{1/7})$/$\tilde O(n^{1/8})$ for random features/Nystrom features, we could achieve a optimal bound for the evaluation error of each player’s Q-function. Generally speaking, our SDEPO algorithm requires a good feature generation method to represent the environment and future work will explore alternative  feature generation methods such as divide-and-conquer approaches [2] and greedy basis selection techniques [3].
>
> Response to Weakness 3:
>
> Thank you for your insightful suggestion to include more complex and diverse scenarios in our experimental section. Actually, the intention of this paper is to propose a provably efficient method for two-player zero-sum stochastic Markov games with an infinite state space, and our SDEPO achieves the best-known iteration complexity $\tilde O(\frac{1}{(1-\gamma)^3 \epsilon})$. We note that all the existing methods [4,5,6,7,8,9] for two-player zero-sum stochastic Markov games with an infinite state space mainly focus on the theoretical aspect and do not provide any experimental results at all. Actually, these methods all involve a computational inefficient subroutine, i.e., [4,5,6,7] need to solve a difficult ’find_ne’/’find_cce’ subroutine, and [8,9] have to tackle a comprehensive constrained optimization problem.
> With a purpose to explore the ability of our proposed method in handling real-world complex tasks, we derived a practical variant of SDEPO, named SDEPO-NN to deal with both continuous action spaces and continuous state spaces, and direct validated the effectiveness of SDEPO-NN on the simple push task.
>
> To investigate the effectiveness of SDEPO, we conduct a simulated experiment on a simple zero-sum Markov game. (please see Global Response)
>
> We would include more complex and diverse scenarios in the experimental study and add more comparison in our long version of this paper.
>
> [1] Chengzhuo Ni, et al. Representation learning for general-sum low-rank markov games. arXiv preprint arXiv:2210.16976, 2022.
>
> [2] C.-J. Hsieh, et al. A divide-and-conquer solver for kernel support vector machines. In International Conference on Machine Learning, 2014.
>
> [3] A. J. Smola and B. Scholkopf. Sparse greedy matrix approximation for machine learning. In ¨ Pro ceedings of the International Conference on Machine Learning, pages 911–918, San Francisco, 2000. Morgan Kaufmann Publishers.
>
> [4] Qiaomin Xie, et al. Learning zero-sum simultaneousmove Markov games using function approximation and correlated equilibrium. In Conference on Learning Theory, pages 3674–3682. PMLR, 2020.
>
> [5] Zixiang Chen, et al. Almost optimal algorithms for two-player zero-sum linear mixture Markov games. In Proceedings of The 33rd International Conference on Algorithmic Learning Theory, volume 167, pages 227–261. PMLR, 2022.
>
> [6] Chris Junchi Li, et al. Learning two-player mixture markov games: Kernel function approximation and correlated equilibrium. arXiv e-prints, pages arXiv–2208, 2022.
>
> [7] Shuang Qiu, et al. On reward-free RL with kernel and neural function approximations: Single-agent MDP and Markov game. In International Conference on Machine Learning, pages 8737–8747. PMLR, 2021.
>
> [8] Baihe Huang, et al. Towards general function approximation in zero-sum Markov games. In International Conference on Learning Representations, 2022.
>
> [9] Chi Jin, et al. The power of exploiter: Provable multi-agent RL in large state spaces. In International Conference on Machine Learning, pages 10251–10279. PMLR, 2022.

---

> > ### Author Response · Authors · 2024-08-14
> >
> > We thank you for your review and appreciate your time reviewing our paper. The end of the discussion period is close. We would be grateful if we could hear your feedback regarding our answers to the reviews. We are happy to address any remaining points during the remaining discussion period. Thanks a lot in advance!

---

### Author Rebuttal · Authors · 2024-08-07

We thank all of the reviewers for the detailed comments. Here we numerically verified the convergence of SDEPO. Here, we designed a simple zero-sum Markov game with a continuous state and finite action space ($\mathcal{S} = \mathbb{R}$,$ |\mathcal{A}|$ = 5). As for the transition probability and reward function, we segment state space into 42 distinct intervals as follows:

1、One interval from $-\infty$ to -10,

2、40 intervals from [−10,10), divided every 0.5 units,

3、One interval from 10 to $+\infty$.

In the $ith$ interval, $P(s,a,b) = f(s,a,b)+\epsilon$ where $\epsilon\sim\mathcal{N}(0,1)$, $f(s,a,b)=\epsilon_{i,a,b)}$ with $\epsilon_{i,a,b)}\sim Unif(-10.5,10.5)$.
The reward function is $r(s,a,b)=\epsilon_{i,a,b)}^\prime$ with $\epsilon_{i,a,b)^\prime}\sim Unif(-1,1)$ and initial distribution is $Unif(-10.5,10.5)$.

We ran SDEPO for 120 iterations, and measured the convergence of $\underline{\pi}$ by metrics in Proposition 1. As shown in Figure 1, SDEPO with random features and Nyström features both converge after 60 iterations.
We discretized the state space of this environment and compared it with OFTRL [1], a tabular method where the environment is known. We adopted the parameter settings recommended in [1] and adjusted the environment to a 100-horizon setting. As shown in Figure 1 in the pdf file, our method demonstrated superior convergence in this environment. This likely stems from the fact that OFTRL operates on the discretized state space, whereas our method computes on the original state space.

Note that we do not include any algorithm for two-player zero-sum stochastic Markov games with an infinite state space, as all the existing methods [2,3,4,5,6,7] mainly focus on the theoretical aspect and do not provide any experimental results at all. Actually, these methods all involve a computational inefficient subroutine, i.e., [2,3,4,5] need to solve a difficult ’find_ne’/’find_cce’ subroutine, and [6,7] have to tackle a comprehensive constrained optimization problem.

[1] Policy Optimization for Markov Games: Unified Framework and Faster Convergence

[2] Qiaomin Xie, et al. Learning zero-sum simultaneous-move Markov games using function approximation and correlated equilibrium. In Conference on Learning Theory, pages 3674–3682. PMLR, 2020.

[3] Zixiang Chen, et al. Almost optimal algorithms for two-player zero-sum linear mixture Markov games. In Proceedings of The 33rd International Conference on Algorithmic Learning Theory, volume 167, pages 227–261. PMLR, 2022.

[4] Chris Junchi Li, et al. Learning two-player mixture markov games: Kernel function approximation and correlated equilibrium. arXiv e-prints, pages arXiv–2208, 2022.

[5] Shuang Qiu, et al. On reward-free RL with kernel and neural function approximations: Single-agent MDP and Markov game. In International Conference on Machine Learning, pages 8737–8747. PMLR, 2021.

[6] Baihe Huang, et al. Towards general function approximation in zero-sum Markov games. In International Conference on Learning Representations, 2022.

[7] Chi Jin, et al. The power of exploiter: Provable multi-agent RL in large state spaces. In International Conference on Machine Learning, pages 10251–10279. PMLR, 2022.

---

### Decision · Program_Chairs · 2024-09-25

**Decision:**

Accept (poster)

**Comment:**

The paper introduces the Spectral Dynamic Embedding Policy Optimization (SDEPO) algorithm to address two-player zero-sum Markov games with continuous state spaces, providing both theoretical guarantees and practical performance improvements. The approach leverages spectral embedding and offers a convergence rate that matches the best-known single-agent RL results, with a clear motivation and solid theoretical foundation. Overall, the paper is considered technically solid with the recommendation leaning towards acceptance.